# $CO_2$ electrolysis to multi-carbon products in strong acid at ampere-current levels on La-Cu spheres with channels

Jiaqi Feng[1,2], Limin Wu[1,3], Xinning Song[1,3], Libing Zhang [1,3], Shunhan Jia [1,3], Xiaodong Ma[1,3], Xingxing Tan[1,3], Xinchen Kang [1,3], Qinggong Zhu [1,3], Xiaofu Sun [1,3] ✉ & Buxing Han [1,3,4] ✉

Achieving satisfactory multi-carbon ($C_{2+}$) products selectivity and current density under acidic condition is a key issue for practical application of electrochemical $CO_2$ reduction reaction ($CO_2$RR), but is challenging. Herein, we demonstrate that combining microenvironment modulation by porous channel structure and intrinsic catalytic activity enhancement via doping effect could promote efficient $CO_2$RR toward $C_{2+}$ products in acidic electrolyte (pH ≤ 1). The La-doped Cu hollow sphere with channels exhibits a $C_{2+}$ products Faradaic efficiency (FE) of 86.2% with a partial current density of $-775.8$ mA cm$^{-2}$. $CO_2$ single-pass conversion efficiency for $C_{2+}$ products can reach 52.8% at $-900$ mA cm$^{-2}$. Moreover, the catalyst still maintains a high $C_{2+}$ FE of 81.3% at $-1$ A cm$^{-2}$. The channel structure plays a crucial role in accumulating K$^+$ and OH$^-$ species near the catalyst surface and within the channels, which effectively suppresses the undesired hydrogen evolution and promotes C−C coupling. Additionally, the La doping enhances the generation of *CO intermediate, and also facilitates $C_{2+}$ products formation.

Electrocatalytic $CO_2$ reduction reaction ($CO_2$RR) powered by renewable electricity has emerged as a promising strategy in addressing the challenges of climate change and driving the development of a circular carbon economy[1–3]. Among various $CO_2$RR products, multi-carbon ($C_{2+}$) chemicals, such as ethylene ($C_2H_4$) and ethanol ($C_2H_5OH$), have garnered significant attention due to their potential to tap into existing large markets[4–7]. Currently, most research efforts are focused on the design of novel copper-based catalysts to enhance the production activity of $C_{2+}$ products in alkaline or neutral electrolyte systems[8,9], in which an alkaline environment formed around the catalyst surface favors the activation of $CO_2$ molecules and the suppression of the competitive hydrogen evolution reaction (HER), facilitating C−C coupling to $C_{2+}$ products[10–12]. However, $CO_2$ can easily react with local/bulk

OH$^-$ species to produce (bi)carbonate ($CO_3^{2-}$ or $HCO_3^-$), which crosses the anion exchange membrane (AEM) to the anode and then is converted back to $CO_2$ in the anode tail gas, resulting in a low $CO_2$ single-pass conversion efficiency (SPCE, <25%)[13–15]. Moreover, the carbonate formation can also block gas diffusion electrode (GDE) and increase the cell resistance. These challenges significantly impede the potential industrial application of $CO_2$ electrolysis.

Alternatively, electrocatalytic $CO_2$-to-$C_{2+}$ products in acidic system can solve the above problems well. Although OH$^-$ species are also locally produced around the cathode surface in the acidic electrolyte, the generated carbonate would be converted back into $CO_2$ molecules by H$^+$ when it diffuses to the bulk electrode[16–18]. Nevertheless, it is challenging to obtain high $C_{2+}$ products Faradaic efficiency (FE), due to

[1]Beijing National Laboratory for Molecular Sciences, Key Laboratory of Colloid and Interface and Thermodynamics, Center for Carbon Neutral Chemistry, Institute of Chemistry, Chinese Academy of Sciences, Beijing 100190, China. [2]College of Chemical Engineering and Environment, China University of Petroleum (Beijing), Beijing 102249, China. [3]School of Chemical Sciences, University of Chinese Academy of Sciences, Beijing 100049, China. [4]Shanghai Key Laboratory of Green Chemistry and Chemical Processes, State Key Laboratory of Petroleum Molecular & Process Engineering, School of Chemistry and Molecular Engineering, East China Normal University, Shanghai 200062, China. ✉e-mail: sunxiaofu@iccas.ac.cn; hanbx@iccas.ac.cn

the kinetic preference of the acidic environment towards the HER. It was reported that the presence of $K^+$ in acid electrolyte could shield the electric field around cathode and impede the migration of $H^+$ to the catalyst surface, thereby inhibiting the HER[19,20]. On the other hand, slowing the diffusion of generated $OH^-$ away from the catalyst surface can create a high alkaline microenvironment, which facilitated C−C coupling[21–23]. However, due to the limitation of $K^+$ solubility in aqueous electrolytes and the absence of effective strategies to slow $OH^-$ diffusion, the $C_{2+}$ FE and current density in acidic systems are far below those in alkaline/neutral conditions.

The catalyst structure affects the local microenvironment, which determines the thermodynamic and kinetic catalytic processes and has great potential to control the reaction rate, selectivity, stability, etc[24,25]. In particular, the catalyst with the porous channel structure provides a confined space for the reaction, which can tune the diffusion behavior of the reactive/non-reactive species and influence the accessibility of the reactant to the active site. It has been reported that the porous channel structure can increased the surface $OH^-$ concentration, enrich the reactant and thus promoting catalytic performance[26,27]. Additionally, enhancing the intrinsic activity of Cu-based catalyst for $CO_2RR$ is also vital. Doping another metal element into Cu-based catalyst has been demonstrated as an effective method for tuning the binding strength of intermediates and thus enhancing the intrinsic activity of electrocatalytic $CO_2$-to-$C_{2+}$ products[28,29]. Different from d-block metal elements, lanthanide metal (LM) elements possess intense spin-orbit coupling and lanthanide contraction effects. These properties lead to the accumulation of localized electronic states and hold the potential to alter the electronic structure of doped d-block metal species, thus enabling catalytic enhancement[30,31]. Therefore, it can be anticipated that designing a LM-doped Cu-based catalyst with the porous channel structure to enrich $K^+$ and form a high alkaline microenvironment would significantly enhance the performance of $CO_2RR$ to $C_{2+}$ products in an acidic system.

Herein, we have synthesized the La-doped Cu hollow sphere (La-Cu HS) catalyst with the porous channel structure in shell through a facile two-step approach, which exhibits highly efficient $CO_2RR$ to $C_{2+}$ products in strongly acidic electrolyte (pH ≤ 1). A high $C_{2+}$ products FE of 86.2% is obtained at the current density of −900 mA cm$^{-2}$, exceeding the performances of the reported electrocatalysts in acidic system. Particularly, the $C_{2+}$ products FE could be kept over 81% even the current density reached −1 A cm$^{-2}$. The simulations and experiments confirm that the porous channel structure of the La-Cu HS efficiently enriches $K^+$ at the catalyst surface, and easily creates a high alkaline microenvironment in the electrolyte near surface and within the channel of the catalysts, which could suppress HER and facilitate the C−C coupling process. Additionally, in-situ experiments and density function theory (DFT) calculations indicates that the La-O-Cu site formed by doping with La facilitates *CO generation and C−C coupling process. Therefore, combining the advantage of the porous channel structure and La doping effect, the as-fabricated La-Cu HS catalyst exhibits outstanding $CO_2$-to-$C_{2+}$ products performance in acidic electrolyte.

## Results

### Catalysts synthesis and characterization

The La-Cu HS catalyst was synthesized through a two-step process and a hollow sphere structure with channels in shell was successfully obtained through inside-out Ostwald ripening mechanism (Fig. 1A)[32,33]. In brief, La-doped $Cu_2O$ hollow sphere (La-$Cu_2O$ HS) was first obtained via solvothermal method in a Teflon-lined autoclave. Subsequently, the as-prepared La-$Cu_2O$ HS underwent electrochemical reduction process and in-situ converted into La-Cu HS in a flow cell equipped with a gas diffusion electrode (GDE). Scanning electron microscope (SEM) and transmission electron microscopy (TEM) images (Figs. S1, S2) showed that the La-$Cu_2O$ HS possessed a hollow sphere morphology

with an average diameter of about 300 nm, and the shell had obvious porous channel structure. Nitrogen adsorption-desorption isotherm experiment confirmed that La-$Cu_2O$ HS had mesoporous structure and the primary pore size was around 5 nm (Fig. S3). The clear lattice spacing of 0.246 nm corresponding to the lattice plane distance of $Cu_2O(111)$ facet was observed in the high-resolution TEM (HRTEM) image and the powder X-ray diffraction (XRD) patterns (Fig. S4) and X-ray photoelectron spectroscopy (XPS) spectra (Fig. S5) also suggested that the Cu species in the La-$Cu_2O$ HS were $Cu_2O$. In addition, the energy-dispersive X-ray spectroscopy (EDS) mappings indicated that the Cu, La, and O elements are uniformly distributed in La-$Cu_2O$ HS (Fig. S6).

After the electrochemical reduction process, no obvious morphology change was observed from SEM and TEM images (Fig. 1B, C). The spacing of the lattice fringe was 0.208 nm, which can be assigned to the Cu(111) facet (Fig. 1D). The EDS mappings depicted in Fig. 1E revealed a consistent distribution of Cu and La elements, suggesting a uniform doping of La in the entire structure. The La content, determined through EDS, was 1.22 wt%, which closely aligned with 1.03 wt% obtained via inductively coupled plasma optical emission spectroscopy (ICP-OES). For comparison, La-doped Cu with solid sphere morphology (without channels, La-Cu SS) and the Cu catalyst with hollow sphere morphology (without La, Cu HS) were also prepared through similar method (Figs. S7, S8). XRD patterns also proved the transformation of $Cu_2O$ to Cu after the electrochemical reduction process, while no peaks related to La species were observed (Fig. 1F). The La 3d$_{5/2}$ XPS spectra of La-Cu HS and La-Cu SS were displayed in Fig. S9, the La 3d$_{5/2}$ region has well separated spin-orbit components. The binding energy of the La 3d$_{5/2}$ peak for La-Cu HS and La-Cu SS is 835.0 eV, indicating that the La species is in oxidized state. Additionally, the La 3d$_{5/2}$ split, denoted as ΔE, was marked in Fig. S9. It can be observed that La-Cu HS and La-Cu SS show the similar ΔE, which is smaller than that of $La_2O_3$ and $La(OH)_3$, suggesting that the La species oxidized state of La-Cu HS and La-Cu SS is distinct from that of $La_2O_3$ and $La(OH)_3$[34,35]. The Cu Auger LMM spectra demonstrated that Cu mainly appeared as metallic state in all the three samples (Fig. S10). Notably, the binding energy of Cu 2p$_{3/2}$ peak of La-Cu HS and La-Cu SS shifted to lower binding energy comparing to that of Cu HS (Fig. 1G), which was attributed to the charge transfer from La to Cu. This phenomenon can be explained by the higher electronegativity of Cu (1.90) than that of La (1.10). Subsequently, we performed the in-situ X-ray absorption spectroscopy (XAS) measurements (Fig. S11). The Cu K-edge X-ray absorption near-edge spectroscopy (XANES) of La-Cu HS, La-Cu SS, and Cu HS were similar with that of Cu foil (Fig. 1H), suggesting that the Cu chemical state in the three catalysts was $Cu^0$, which was consistent with the XPS analysis. In addition, obvious Cu-Cu coordination at 2.2 Å could be observed in the Fourier transform extended X-ray absorption fine structure spectra (FT EXAFS) of La-Cu HS, La-Cu SS, and Cu HS (Fig. S12), further confirming that Cu existed as metallic state in all the three catalysts. Unfortunately, we failed to obtain effective La XAS signals under operando conditions, due to low La content in the catalysts. We can only determine the La species in La-Cu HS and La-Cu SS by immediately collecting XAS data after $CO_2RR$ testing. The spectra of $La(OH)_3$ and $La_2O_3$ were collected as reference samples for comparison. As revealed by La L$_3$-edge XANES spectra (Fig. S13), the white line intensity of La-Cu HS and La-Cu SS was similar but lower than that of $La(OH)_3$ and $La_2O_3$. It suggested that the La species in La-Cu HS and La-Cu SS was similar, but it was different from those in $La(OH)_3$ and $La_2O_3$. One main peak at around 2.0 Å was observed in the FT EXAFS spectra of La-Cu HS and La-Cu SS, which was attributed to La-O coordination. There is no La-La coordination peak at around 4.0 Å in La-Cu HS and La-Cu SS, confirming that the La species exists as single atomic species in La-Cu HS and La-Cu SS without long-range coordination to other La centers. Moreover, there is a peak at around 4.5 Å in La-Cu HS and La-Cu SS FT EXAFS spectra, which could

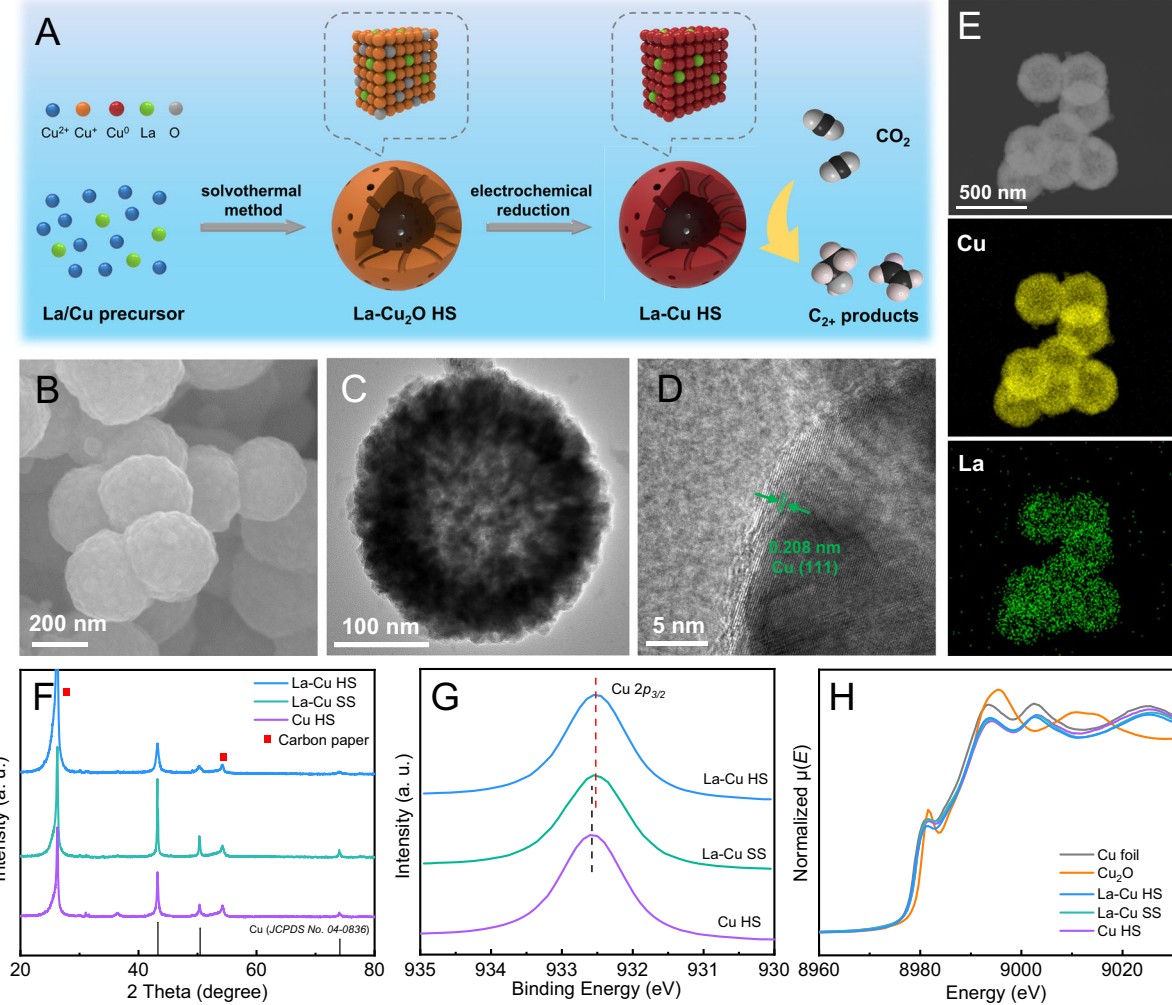

**Fig. 1 | Morphology and structural characterization. A** Schematic illustration for fabrication of La-Cu HS. **B** SEM, (**C**) TEM, and (**D**) HRTEM images of La-Cu HS. **E** EDS mappings of La-Cu HS (yellow and green represent Cu and La elements, respectively). **F** XRD patterns and (**G**) Cu 2p$_{3/2}$ XPS spectra of La-Cu HS, La-Cu SS, and Cu HS. **H** In-situ Cu K-edge XANES spectra of La-Cu HS, La-Cu SS, Cu HS, and the reference samples.

be assigned to the La-O-Cu scattering[36]. Therefore, the La species in La-Cu HS and La-Cu SS are the form of La-O$_x$ sites on the Cu surface.

## Simulation and experiment studies

To investigate the contribution of porous channel structure to enriching K$^+$ concentration and enhancing local pH, we performed a comparative study of hollow sphere with porous channel structure in shell and solid sphere structure without channels by COMSOL Multi-physics finite-element-based simulations. The bulk K$^+$ concentration was set at 3 M. We simulated the K$^+$ concentration distribution over the surface of hollow sphere with channels, solid sphere, and within channels of the shell under various current. The results indicated that the K$^+$ concentration on the surface of hollow sphere with channels and solid sphere increases with the increasing current (Fig. S14 and Fig. 2A, B), while the surface of hollow sphere with channels showed much higher K$^+$ concentration than solid sphere surface under the same current. In the interior of the channels, the K$^+$ concentration was higher near the outside surface and decreases gradually from outside to inside. Meanwhile, we also simulated the negative charge distribution over the hollow sphere with porous channel structure in shell and solid sphere structure without channels by COMSOL. Fig. S15 showed a pronounced localized enrichment of charge on the surface of the hollow sphere under all current conditions, with a significantly higher charge density was observed in the vicinity of the channels. The

accumulation of negative charges near the channels could attract K$^+$, facilitating the diffusion of K$^+$ into the channels. To further verify the ability of hollow sphere structure to enrich K$^+$, we measured the K 2p XPS spectra over the GDE loaded with La-Cu HS and La-Cu SS, after CO$_2$RR under the same conditions (Fig. 2C). It is worth noting that no peaks corresponding to KHCO$_3$ precipitate was found in the XRD patterns of the electrode after CO$_2$ reduction (Fig. S16), and the pH of the electrolyte after CO$_2$RR was 0.9, which could exclude the presence of KHCO$_3$ precipitate on the GDE surface. The XPS results showed a significant increase of K$^+$ content on the GDE loaded with La-Cu HS compared with that loaded with La-Cu SS, experimentally proving the conclusion of the simulation results. The elevated concentration of K$^+$ is conducive to inhibiting HER.

On the other hand, we also simulated pH value of the electrolyte near the surface of hollow sphere and solid sphere as well as inside the channels in shell of hollow sphere at different current. The bulk electrolyte pH value was set as 1. As shown in Fig. S17, hollow sphere can form an alkaline microenvironment near the surface at lower current. At the current density of −700 mA, high alkaline microenvironment was generated near hollow sphere surface with channels, and the maximum pH value can reach 13.7, while that of solid sphere is only 11.8 (Fig. 2D). The phenomenon was more obvious at −900 mA, and the pH value can keep over 13 even at a distance of 300 nm away from the hollow sphere surface with channels (Fig. 2E). Moreover, the inside of

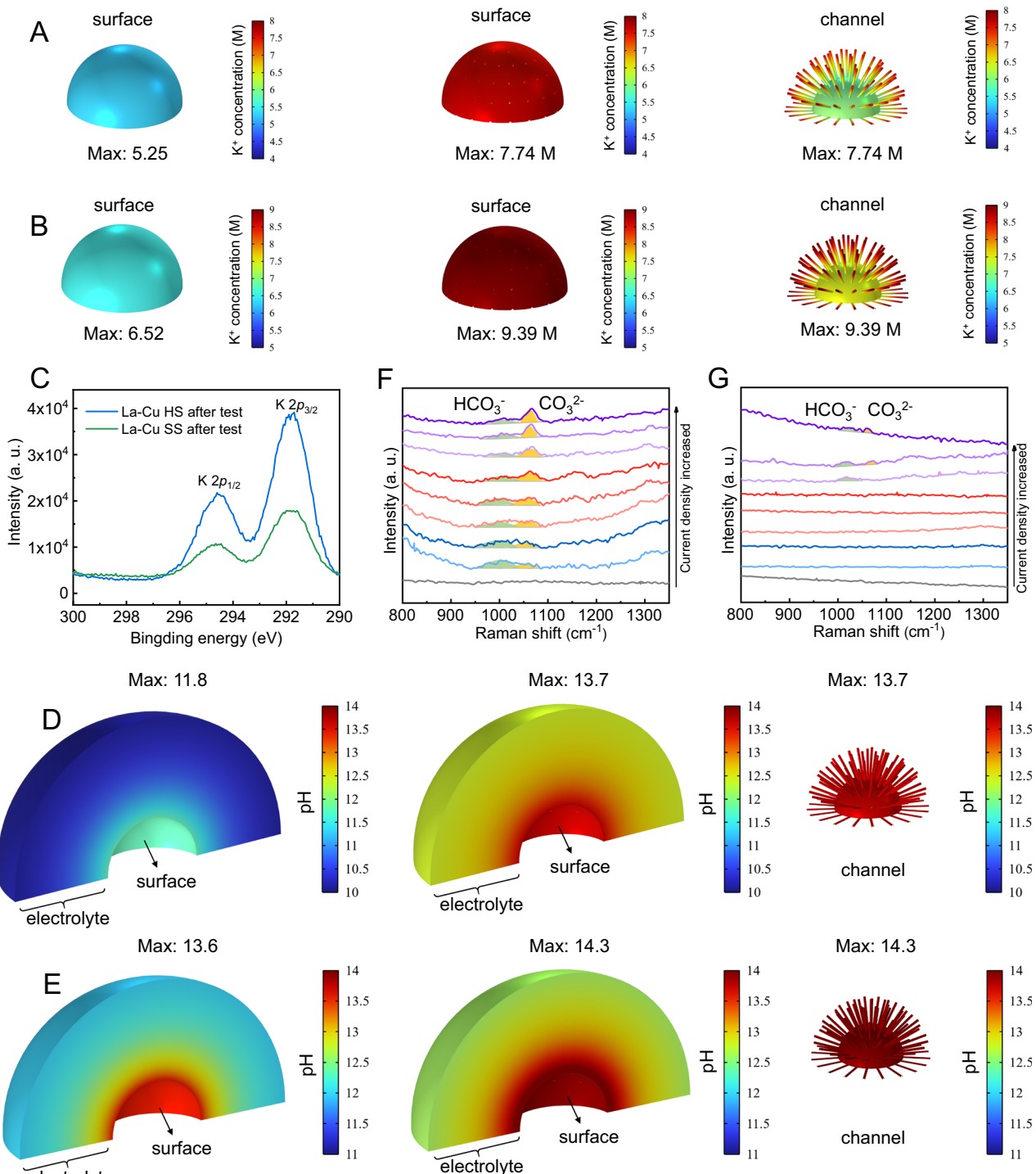

**Fig. 2 | Simulation and experiment studies. A** From left to right are the K⁺ concentration distribution over surface of solid sphere, hollow sphere with channels, and within channels of hollow sphere at −700 mA. **B** From left to right are the distribution of K⁺ concentration over surface of solid sphere, hollow sphere, and within channels of hollow sphere at −900 mA. **C** K 2p XPS spectra of La-Cu HS and La-Cu SS after CO₂RR testing in 0.05 M H₂SO₄ containing 3 M KCl electrolyte.

**D** From left to right are the pH distribution in electrolyte near surface of solid sphere, hollow sphere, and within channels of hollow sphere at −700 mA. **E** From left to right are the pH distribution in electrolyte near surface of solid sphere, hollow sphere, and within channels of hollow sphere at −900 mA. In-situ SERS spectra of (**F**) La-Cu HS and (**G**) La-Cu SS under different current densities in 0.05 M H₂SO₄ containing 3 M KCl electrolyte.

the channels can also show a high alkaline environment when current is larger than −700 mA. It has been reported that the equilibrium reaction between generated $HCO_3^-$ and $CO_3^{2-}$ species can reflect the local pH near the catalyst surface, and an alkaline microenvironment promotes the forward movement of the reaction equilibrium towards $CO_3^{2-}$ formation[37,38]. Therefore, we carried out in-situ surface-

enhanced Raman spectroscopy (SERS) to monitor the generated $HCO_3^-$ (1011 cm⁻¹) and $CO_3^{2-}$ (1063 cm⁻¹) species over La-Cu HS and La-Cu SS in 0.05 M H₂SO₄ containing 3 M KCl electrolyte under different current densities (Fig. S18). As displayed in Fig. 2F, obvious $HCO_3^-$ and $CO_3^{2-}$ peaks were all detected over La-Cu HS at different current densities, indicating that generation of OH⁻ created the alkaline

microenvironment. As the current density increased, the peak for $CO_3^{2-}$ became more prominent while the peak for $HCO_3^-$ became weaker, demonstrating that the gradually higher alkaline microenvironment was formed. However, the $HCO_3^-$ and $CO_3^{2-}$ peaks over La-Cu SS spectra could be only observed at large current density (Fig. 2G), and the area ratio of $CO_3^{2-}$ to $HCO_3^-$ was much lower than that of La-Cu HS. Thereby, the in-situ SERS results provided experimental evidence that hollow sphere with porous channel structure facilitates the formation of a highly alkaline microenvironment near the catalyst surface, which may play a crucial role in inhibiting HER and promoting C−C coupling reaction.

**Electrocatalytic CO₂RR performance**

A flow cell equipped with GDE was used to assess the CO₂RR performance of La-Cu HS, La-Cu SS, and Cu HS under different current densities, and 0.05 M $H_2SO_4$ aqueous solution containing 3 M KCl was used as catholyte (pH = 0.87)[21]. CO, CH₄, C₂H₄ and H₂ were detected in the gas-phase by gas chromatography, and formate, ethanol, acetate and n-propanol were detected in liquid-phase products through ¹H nuclear magnetic resonance spectroscopy[39,40]. Figure 3A displays the C₂₊ products FE (FE$_{C2+}$) over La-Cu HS, La-Cu SS, and Cu HS at the current densities ranging from −300 to −1000 mA cm⁻². The La-Cu HS exhibited a FE$_{C2+}$ over 75% in the full current density range, and the

maximum FE$_{C2+}$ reached 86.2% at −900 mA cm⁻², corresponding to the C₂₊ products partial current density ($j_{C2+}$) as high as −775.8 mA cm⁻² (Fig. 3B). More importantly, it still maintained a high FE$_{C2+}$ of 81.3% at −1000 mA cm⁻², and $j_{C2+}$ could reached −813.0 mA cm⁻². The performance of La-Cu HS was much better than that of La-Cu SS, and Cu HS. The H₂ FE of La-Cu HS was suppressed within 10% in the whole current density range (Fig. S19), while it exceeded 20% over La-Cu SS and Cu HS at high current density. Figure 3C reveals a high C₂₊-to-C₁ ratio of 9.1 on La-Cu HS, which is 2.0-fold and 4.3-fold greater than that on La-Cu SS and Cu HS, respectively. In addition to La, the influence of doping other LM elements into Cu on CO₂RR-to-C₂₊ products were also investigated. We doped Pr, Tb, and Er into Cu using the same method as La-Cu HS and tested their CO₂RR activity under identical electrolyte and current density conditions. The products distribution was presented in Fig. S20, and the results indicated that La-Cu exhibited highest C₂₊ products FE.

Based on the results above, we can find that both the channels within the shell and La doping promoted the C₂₊ products selectivity. To investigate the influence of La dopant on the CO₂RR performance, we prepared a series of La-Cu HS catalysts with varying La content through changing La/Cu feed ratios (the molar ratios of La/Cu = 0.1, 0.2, 0.3, and 0.4), and the La contents determined by ICP-OES were 0.78%, 0.83%, 1.03%, and 1.39%, respectively (Figs. S21–S23). The

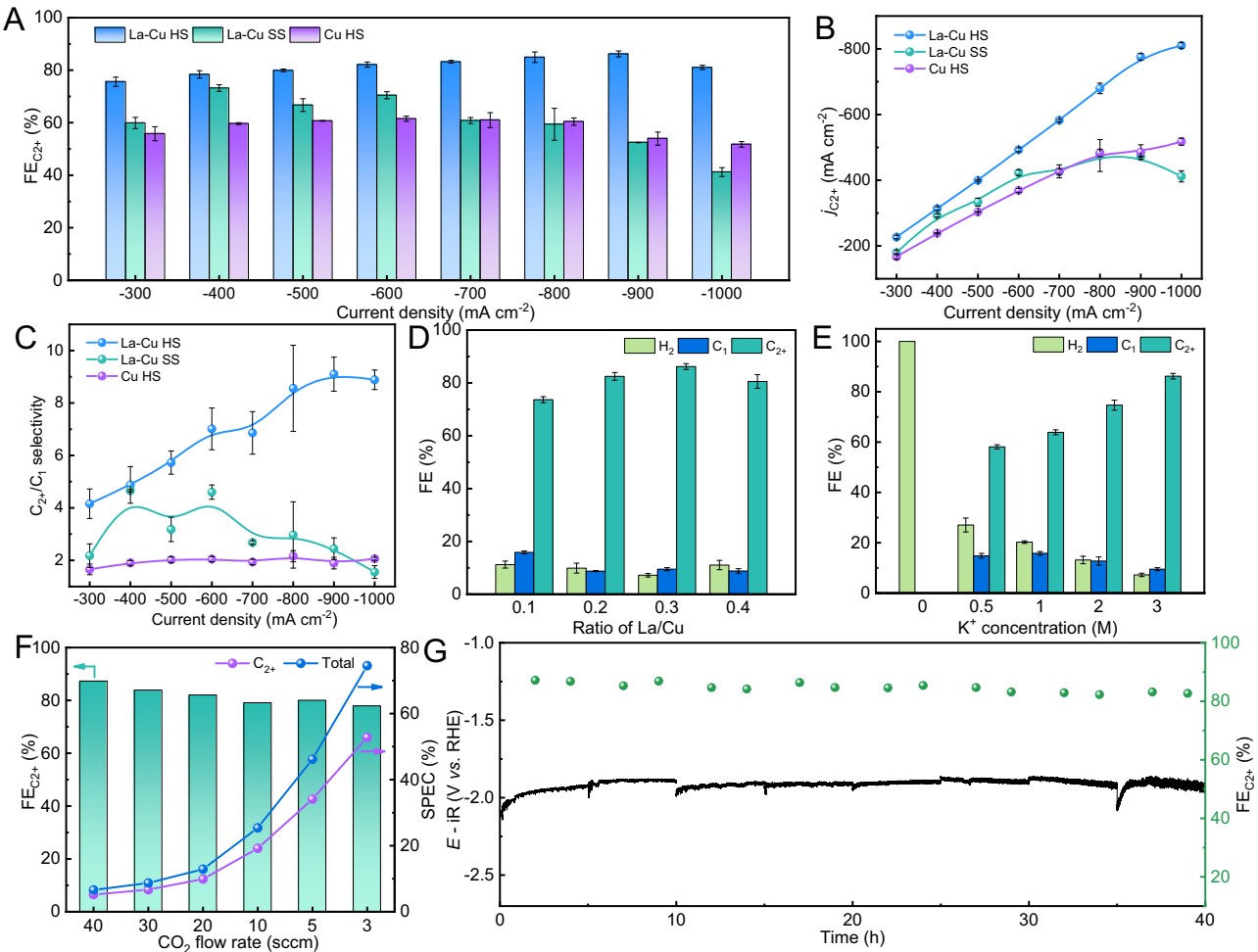

**Fig. 3 | Electrocatalytic CO₂RR performance. A** The FEs for C₂₊ products, (**B**) C₂₊ products partial current density and (**C**) C₂₊/C₁ selectivity over La-Cu HS, La-Cu SS, and Cu HS under different current densities. **D** The products FE over La-Cu HS with different La/Cu ratio at −900 mA cm⁻². **E** The products FE over La-Cu HS in 0.05 M $H_2SO_4$ aqueous solution containing different KCl concentrations at −900 mA cm⁻².

Values are means and error bars indicate s.d. (n = 3 replicates). **F** C₂₊ products FE, SPCE and total products SPCE of La-Cu HS under different CO₂ gas flow rate at −900 mA cm⁻². **G** Long-term stability of La-Cu HS at a constant current density of −900 mA cm⁻² (The electrode was washed, then dried and the electrolyte was refreshed at intervals 5 h).

$CO_2$RR performance of the La-Cu HS with various La doping content at $-900$ mA cm$^{-2}$ is presented in Fig. 3D and Fig. S24, and a volcano-shaped relationship can be observed between $C_{2+}$ products FE and La doping content. The maximum $C_{2+}$ products FE (86.2%) was obtained when La/Cu ratio is 0.3, confirming that the appropriate doping of La into Cu would enhance $C_{2+}$ products selectivity. Comparing with state-of-the-art electrocatalysts used in acidic $CO_2$RR systems, the La-Cu HS exhibited much higher FE and partial current density for the generation of $C_{2+}$ products (Table S1).

The influence of K$^+$ concentration on $CO_2$RR performance of La-Cu HS was also studied. Only H$_2$ was detected when 0.05 M H$_2$SO$_4$ aqueous solution was used as catholyte (Fig. 3E, Fig. S25). The H$_2$ FE gradually decreased with an increase of K$^+$ concentration in 0.05 M H$_2$SO$_4$ aqueous solution, and was as low as 7.2% at 3 M K$^+$. Meanwhile, the significant improvement in $C_{2+}$ products selectivity (58.1%) was obtained in 0.5 M K$^+$, and $C_{2+}$ products FE obviously increased as K$^+$ concentration further increased. Overall, the results above suggest that high K$^+$ concentration is beneficial for suppressing HER and promoting $C_{2+}$ products selectivity in an acidic electrolyte.

Additionally, the utilization of acidic system could minimize carbonate formation, and thus is advantageous in overcoming carbon utilization limitations observed in neutral and alkaline solutions. Upon reducing the flow rate of $CO_2$ from 40 to 3 standard cubic centimeters per minute (sccm), the SPCE for all the products over La-Cu HS increased from 6.6% to 74.5% at $-900$ mA cm$^{-2}$, and the SPCE for $C_{2+}$ products could reach 52.8% at 3 sccm (Fig. 3F, Fig. S26). It was high than the reported systems in alkaline/neutral electrolyte, highlighting the advantages of catalytic system. We also evaluated the long-term stability of the La-Cu HS in acidic system at $-900$ mA cm$^{-2}$. To alleviate the problem of the gradual loss of hydrophobicity of GDE, the $CO_2$RR was interrupted every 5 h and the GDE were removed, washed with deionized water and followed by dryness under N$_2$ atmosphere. As shown Fig. 3G, no significant decay of the measured potential was observed and the FE$_{C2+}$ remained above 80% after operation for 40 h, implying the good durability. The morphology and structure of the La-

Cu HS were well preserved after stability test (Fig. S27). Furthermore, the EDS mapping confirmed that La element still existed on the surface of La-Cu HS after $CO_2$RR (Fig. S28), and the La content was 1.18 wt%, similar to the La content of La-Cu HS before $CO_2$RR. The XPS confirmed that no notable changes were observed over element composition after electrolysis (Fig. S29). The result of ICP-OES indicated that the La content in the electrolyte after $CO_2$RR can be neglected, confirming that atomic La species in the catalyst were stable during $CO_2$RR. Taken together, the La-Cu HS exhibited great promise for applications involving the conversion of $CO_2$ into $C_{2+}$ products in acidic system.

## Mechanistic studies

In order to characterize the intermediates during $CO_2$RR, in-situ attenuated total reflection-surface-enhanced IR absorption spectroscopy (ATR-SEIRAS) (Fig. S30) were conducted over La-Cu HS and Cu HS at various applied potentials (Fig. 4A, B). The peaks at around 2120 cm$^{-1}$, corresponding to *CO atop intermediates[41,42], appeared on both La-Cu HS and Cu HS. The peak intensity of *CO over La-Cu HS at $-0.9$ and $-1.0$ V was obviously stronger than that on Cu HS, indicating that La-Cu HS was highly effective in activating and reducing $CO_2$ to *CO. However, the *CO peak intensity in the Cu HS spectrum increased rapidly as the applied potentials became more negative, and it surpassed the intensity observed in the La-Cu HS spectrum within the applied potential range of $-1.1$ to $-1.3$ V. Meanwhile, the appearance of *OCCO peak at 1594 cm$^{-1}$ demonstrated that C–C coupling reaction occurred via *CO dimerization[43]. We integrated the *CO peak and O*CCO peak, and compared the peak area ratio of O*CCO to *CO within the applied potential range of $-1.1$ to $-1.3$ V. Figure 4C shows that the peak area ratio in La-Cu HS spectrum increased gradually as the applied potentials became more negative, which was notably higher than that of Cu HS. It suggested that La-Cu HS facilitated the coupling of *CO into *OCCO at higher applied potentials. In contrast, *CO tended to accumulate on Cu HS surface due to its lower catalytic activity in C–C coupling[44,45]. Therefore, in-situ ATR-SEIRAS experiment results confirmed that the doping of La facilitates *CO generation and C–C

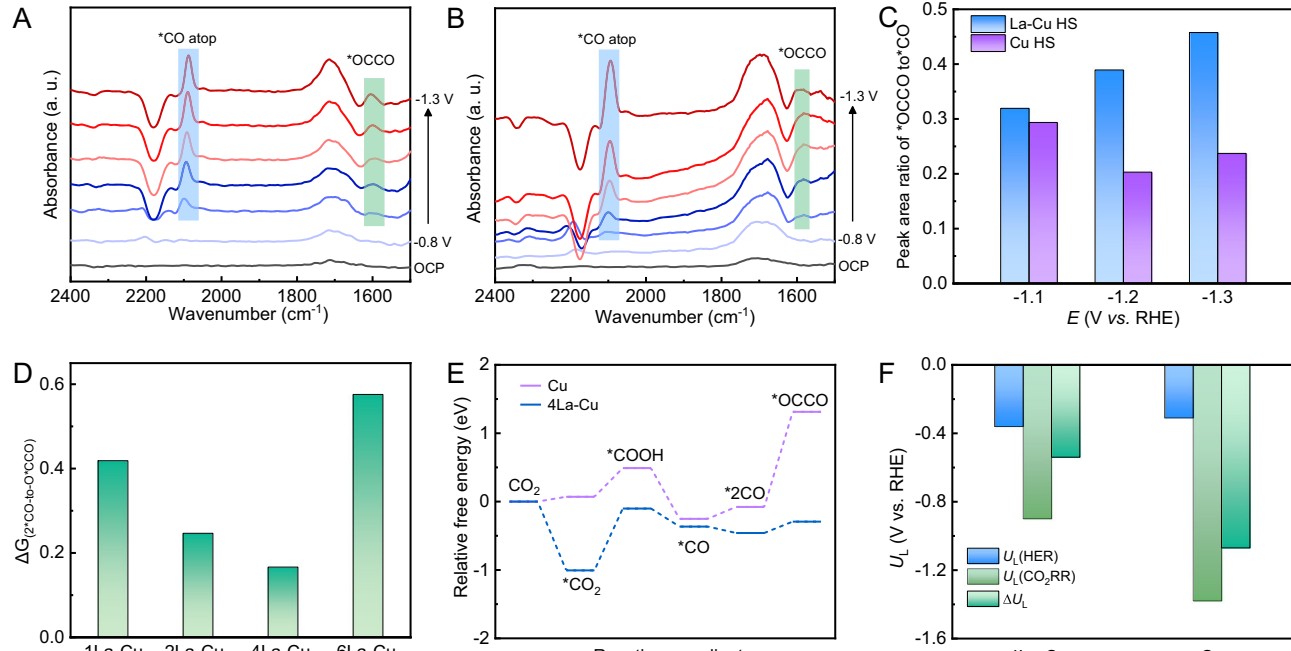

**Fig. 4 | Mechanistic studies.** In-situ ATR-SEIRAS recorded at different potentials for (**A**) La-Cu HS and (**B**) Cu HS during $CO_2$RR. **C** The peak area ratio of *OCCO to *CO as a function of different potentials for La-Cu HS and Cu HS. **D** The relationship between the Gibbs free energy change of the 2*CO-to-O*CCO step and La doping content. **E** The Gibbs free energy diagrams of $CO_2$ reduction to O*CCO on surfaces of 4La-Cu and Cu models, respectively. **F** $U_L$ (HER), $U_L$($CO_2$RR), and $\Delta U_L$ of 4La-Cu and Cu, respectively.

coupling process during $CO_2$-to-$C_{2+}$ products. Subsequently, in-situ XAS experiments were performed to monitor the local structure of Cu during $CO_2$RR under different reaction time. As shown in Fig. S31, the XANES spectrum of La-Cu HS was similar to that of Cu foil, and no obvious difference was observed as $CO_2$RR reaction time increased. Moreover, only peaks corresponding to Cu-Cu coordination were observed in all FT EXAFS spectra, further indicating that the Cu local structure of La-Cu HS kept stable during $CO_2$RR.

To further verify the abovementioned conclusion, we conducted the first-principles calculation based on density functional theory (DFT). La-doped Cu model (La-Cu) was constructed by introducing La and O atoms into Cu(111) crystal plane to form La-O-Cu sites (Fig. S32). In order to investigate the effect of La atom content on the C−C coupling reaction, we introduced 1, 2, 4, and 6 La and O atoms into Cu(111) surface (denoted as 1La-Cu, 2La-Cu, 4La-Cu, 6La-Cu), constructed models with varying La and O atom contents, and calculated the Gibbs free energy change of the 2*CO-to-O*CCO step ($\Delta G_{(2*CO\text{-to-}O*CCO)}$). Notably, to account for the influence of $H_2O$ and $K^+$ on the reaction while simultaneously reducing computational complexity, we simplified the model by placing one $K^+$ hydrated with six water molecules on the surface of the constructed model (Figs. S33–S37)[21,46]. As shown in Fig. 4D, with the increase in the number of La atoms, the $\Delta G_{(2*CO\text{-to-}O*CCO)}$ gradually decreased. When the number of doped La atoms was 4, the $\Delta G_{(2*CO\text{-to-}O*CCO)}$ reached its minimum value of 0.17 eV. Further increasing the number of doped La atoms resulted in the increase of $\Delta G_{(2*CO\text{-to-}O*CCO)}$. The results indicated that moderate of La doping was favorable to C−C coupling process, which aligned with the volcano-shaped relationship observed in the experiments between $C_{2+}$ products FE and La doping content. Additionally, we conducted ab initio molecular dynamics (AIMD) simulations over 4La-Cu under 300 K (Fig. S38), the results confirmed the robust stability of 4La-Cu configuration, where La and O atoms can exist on Cu(111) surface stably.

Figure 4E shows the Gibbs free energy diagrams for $CO_2$ to *OCCO on Cu and 4La-Cu models (Figs. S39, S40). The *$CO_2$ formation was an exothermic process on 4La-Cu, suggesting that the $CO_2$ molecule was favorable to be adsorbed on 4La-Cu surface. The *OCCO formation from 2*CO was the potential limiting step for the Cu model. However, when 4 La atoms were doped onto the Cu surface, the $\Delta G_{(2*CO\text{-to-}O*CCO)}$ decreased from 1.39 eV to 0.17 eV, further demonstrating that the La dopants was favorable to C−C coupling process, and thus improving $C_{2+}$ products selectivity. The HER over Cu and 4La-Cu were calculated (Fig. S41), the energy barrier over 4La-Cu was 0.36 eV, which was larger than that over Cu (0.31 eV), indicating that the doping of La is beneficial for suppressing HER. Additionally, we adopted the difference between the thermodynamic limiting potentials ($\Delta U_L$) for $CO_2$-to-*OCCO and HER (i.e., $U_L(CO_2RR)$-$U_L(HER)$) to compare the selectivity of $CO_2$RR and HER[47,48], the more positive $\Delta U_L$ corresponds to higher selectivity toward $CO_2$ reduction. The potential limiting step of $CO_2$-to-*OCCO over Cu was 2*CO to O*CCO step, corresponding to $U_L(CO_2RR)$ of −1.39 eV, while it was *$CO_2$ to *COOH step over 4La-Cu, corresponding to $U_L(CO_2RR)$ of −0.90 eV. The $U_L(CO_2RR)$, $U_L(HER)$ and $\Delta U_L$ of 4La-Cu and Cu were displayed in Fig. 4F. The result suggested that 4La-Cu was more favorable to undergoing $CO_2$RR compared to Cu.

## Discussion

In summary, La-Cu HS has been successfully synthesized and verified as the efficient catalyst for $CO_2$RR to $C_{2+}$ products in strongly acidic electrolyte. It exhibited an outstanding $C_{2+}$ products FE of 86.2% with a partial current density of −775.8 mA cm$^{-2}$, a high SPCE of 52.8% and remarkable stability. Moreover, the $C_{2+}$ products FE could be kept over 81% at high current density of −1 A cm$^{-2}$. Detailed studies demonstrated that the channel structure could concentrate the $K^+$ and OH$^-$ species at surface and in channels of the catalyst to suppress HER and promote the C−C coupling process. Meanwhile, the doping of La favored *CO

generation and decreased the energy barrier of C−C coupling reaction, leading to the high activity for $CO_2$-to-$C_{2+}$ products significantly. This work provides an efficient strategy for promoting $CO_2$RR performance in an acidic system by combing microenvironment adjustment and intrinsic catalytic activity enhancement. We believe that it would drive further innovation and development for the conversion and utilization of $CO_2$.

## Methods

### Materials

Copper nitrate trihydrate ($Cu(NO_3)_2\cdot 3H_2O$, purity > 99%), lanthanum nitrite hexahydrate ($La(NO_3)_3\cdot 6H_2O$, purity > 99%), potassium chloride (KCl, purity = 99.8%), phenol (purity > 99.5%), sodium 2, 2-dimethyl-2-silapentane-5-sulfonate (DSS, 99%), deuterium oxide ($D_2O$, purity > 99.9), gas diffusion electrode (GDE, YLS-30), Pt foil, Nafion D-521 dispersion (5 wt%), and proton exchange membrane (N117) were purchased from Alfa Aesar China Co., Ltd. Ethylene glycol (($CH_2OH)_2$, purity = 99%), sulfuric acid ($H_2SO_4$, purity = 98%) were obtained from Alfa Aesar China Co., Ltd. $CO_2$ (99.999%) was provided by Beijing Analytical Instrument Company. All materials were used directly with no further purification.

### Catalyst synthesis

The La doped $Cu_2O$ hollow sphere (La-$Cu_2O$ HS) was fabricated by the solvothermal method, and a recrystallization process (i.e., inside-out Ostwald ripening mechanism) induced the core evacuation, resulting in a hollow sphere structure with channels in shell was successfully obtained without using templates. Firstly, a mixture comprising 400 mg $Cu(NO_3)_2\cdot 3H_2O$ and 215 mg $La(NO_3)_3\cdot 6H_2O$ was added into 20 mL ethylene glycol, and stirred overnight. Then, the above mixture was transferred into 50 mL Teflon-lined stainless-steel autoclave and heated at 160 °C for 6 h in an oil bath under continuous stirring. After naturally cooling down to room temperature, the precipitate obtained was thoroughly washed multiple times with ethanol and deionized water, followed by drying at 60 °C overnight. The La doped Cu hollow sphere (La-Cu HS) was obtained through conducting electrochemical reduction of La-$Cu_2O$ HS within a flow cell. After 1 mg La-$Cu_2O$ HS was dispersed on GDE, a current density of 500 mA cm$^{-2}$ was carried out for a duration of 300 s. The 0.05 M $H_2SO_4$ solution aqueous containing 3 M KCl and 0.1 M $H_2SO_4$ solution aqueous were used as catholyte and anolyte, respectively. The flow rate of $CO_2$ gas was controlled to be 40 standard cubic centimeter per minute (sccm). To synthesize La-Cu HS with varying La content, the same procedure was employed with different La/Cu ratios in the feedstocks.

The La doped Cu solid sphere without channels (La-Cu SS) was synthesized using a similar procedure to that of the La-Cu HS, with the only difference being the temperature adjustment during the solvothermal method, which was set to 130 °C. At lower temperature, nucleation and particle grow rates are slow, which favor the formation of solid structure kinetically based on an unusual insideout Ostwald ripening mechanism. The Cu HS was synthesized following the same procedure as that of La-Cu HS, except that the Teflon-lined stainless-steel autoclave was heated in an oven and without $La(NO_3)_3\cdot 6H_2O$ addition.

### Material characterization

Scanning electron microscopy (SEM) analysis was conducted using a HITACHI S-8020 instrument. Transmission electron microscopy (TEM), high-resolution TEM, and corresponding energy-dispersive spectroscopy (EDS) were performed utilizing a JEOL JEM-2100F instrument. X-ray diffraction (XRD) analysis was carried out using an X-ray diffraction instrument (Model D/MAX2500, Rigaka) with Cu-Kα radiation. The scanning range for 2θ was set from 5° to 90°, with a scanning rate of 5° min$^{-1}$. X-ray photoelectron spectroscopy (XPS) characterization was performed using a Thermo Fisher Scientific ESCA

Lab 250Xi instrument. The XPS measurements employed 200 W monochromatic Al Kα (1486.6 eV) radiation under a pressure of $3 \times 10^{-10}$ mbar. The XPS spectra were calibrated using the C 1 s peak at 284.4 eV. X-ray absorption data at the Cu K-edge of the catalysts were recorded in fluorescence mode at the 1W2B beamline of the Beijing Synchrotron Radiation Facility (BSRF). All collected spectra were analyzed using the Athena and Artemis programs within the IFEFFIT software packages.

## Electrocatalytic $CO_2$ reduction

To prepare the working electrode, a uniform catalyst ink was formulated by combining 1 mg of the synthesized catalyst, 400 μL of isopropanol, and 10 μL of Nafion D-521 dispersion (5 wt%). The mixture was subjected to sonication for 30 min to ensure homogeneity. Subsequently, the catalyst ink was carefully deposited onto the GDE of $0.5 \times 2 \, cm^2$ in area by a micropipette, aiming for a catalyst loading of approximately 1 mg cm$^{-2}$. The electrochemical investigations were carried out using a CHI660E electrochemical workstation equipped with a high current amplifier, CHI680C. A flow cell configuration was utilized, comprising a gas chamber, a cathodic chamber, and an anodic chamber. The working electrode was positioned between the gas chamber and the cathodic chamber, while the counter electrode was placed in the anodic chamber. A proton exchange membrane of $1.5 \times 2.5 \, cm^2$ in area effectively separated the cathodic and anodic chambers. An Ag/AgCl electrode served as the reference electrode, and a Pt foil was employed as the counter electrode. Ohmic resistance (R) of the cell was 1.3 Ω, which was measured by electrochemical workstation. The catholyte, a 0.05 M $H_2SO_4$ solution aqueous with 3 M KCl (the pH value was determined by a calibrated pH meter), and the anolyte, a 0.1 M $H_2SO_4$ solution aqueous, were continuously circulated through the cathodic and anodic chambers, respectively, using peristaltic pumps. The flow rates for the catholyte and anolyte were set at 10 mL min$^{-1}$ and 30 mL min$^{-1}$, respectively. To control the flow of $CO_2$ gas, a mass flow controller was utilized, maintaining a flow rate of 40 mL min$^{-1}$ through the gas chamber.

The gaseous products resulting from the electrochemical reactions were collected using a gas bag and subsequently analyzed using gas chromatography (GC). The GC instrument employed was a GC 7890B, equipped with a thermal conductivity detector (TCD). Argon was utilized as the carrier gas during the analysis. For the analysis of the liquid product, $^1$H nuclear magnetic resonance (NMR) spectroscopy was employed. The NMR measurements were conducted on a Bruker Advance III 400 HD spectrometer using deuteroxide ($D_2O$) as the solvent. The reference compound used for ethanol, acetic acid, and n-propanol was DSS (2,2-dimethyl-2-silapentane-5-sulfonic acid), while phenol served as the reference for formate. The Faradaic efficiency (FE) of products was calculated by the equation:

$$FE = \frac{zFn}{Q} \times 100\% \tag{1}$$

Where $z$ represents the number of electrons transferred for product formation, $n$ is the mole of product obtained from GC/$^1$H NMR, $F$ is Faraday constant (96485 C mol$^{-1}$) and the $Q$ is the amount of cumulative charge recorded by the electrochemical workstation.

The $CO_2$ single-pass conversion efficiency (SPCE) was calculated as following equation:

$$SPCE = \frac{60s \times \sum \left( I \times x_i \times FE_i \div (N_i \times F) \right)}{flow\,rate\,(L/\min) \times 1\,\min \div 24.5\,(L/\min)} \tag{2}$$

Where I is current density, $x_i$ is mole ratio of $CO_2$ to $i$ product, $FE_i$ is the faradaic efficiency of $i$ product, $N_i$ is the number of electron transfer for $i$ product molecule.

## In-situ surface-enhanced Raman spectroscopy (SERS)

The measurement was conducted using a Horiba LabRAM HR Evolution Raman microscope within a modified flow cell setup. The GDE loaded with the catalyst served as the working electrode, while an Ag/AgCl electrode and a Pt wire were utilized as the reference electrode and counter electrode, respectively. In the flow cell, the catholyte consisted of a 0.05 M $H_2SO_4$ solution aqueous with 3 M KCl, while the anolyte comprised a 0.1 M $H_2SO_4$ solution aqueous. These solutions were continuously circulated through the cathodic and anodic chambers using peristaltic pumps operating at a rate of 20 mL min$^{-1}$. A 785 nm laser was employed for the measurements, and the Raman signals were recorded with a 20 s integration time and an averaging of two scans. In-situ SERS was performed by applying various current densities ranging from 5 to 50 mA cm$^{-2}$ to the working electrode. High current densities during in-situ Raman tests would attenuate the SERS signal due to the significant evolution of gas products. It should also be noted that the SERS results are serving as an indication of the interfacial pH in the modified setup.

## In-situ attenuated total reflection-surface-enhanced IR absorption spectroscopy (ATR-SEIRAS)

The measurement was conducted in a modified electrochemical cell, the catalyst was dropped on germanium ATR crystal deposited with Au film, which was used as working electrode, an Ag/AgCl electrode and a Pt wire were utilized as the reference electrode and counter electrode, respectively. The cell was integrated into NICOLET 6700 FTIR spectrometer equipped with MCT detector cooled by liquid nitrogen. A 0.05 M $H_2SO_4$ solution aqueous with 3 M KCl aqueous solution was used as electrolyte and circulated through the cathodic chamber and anodic chamber by peristaltic pumps at a rate of 10 mL min$^{-1}$.

## In-situ XAS measurements

The measurement was performed on a custom-designed flow cell, the catalyst loaded on GDE was used as working electrode, an Ag/AgCl electrode and a Pt wire were utilized as the reference electrode and counter electrode, respectively. The catholyte consisted of a 0.05 M $H_2SO_4$ solution aqueous with 3 M KCl, while the anolyte comprised a 0.1 M $H_2SO_4$ solution aqueous. These solutions were continuously circulated through the cathodic and anodic chambers of the flow cell using peristaltic pumps, operating at a flow rate of 10 mL min$^{-1}$. The data was recorded in fluorescence mode with constantly flowed gaseous $CO_2$. The La $L_3$-edge spectra of the samples were collected by ex-situ method. The catalyst was loaded on GDE and conducted $CO_2$RR test at current density of −900 mA cm$^{-2}$ for 400 s in flow cell, then the GDE was taken out from flow cell and immediately proceed with the data collection of the La $L_3$-edge spectra using fluorescence mode.

## COMSOL multiphysics simulations

A three-dimensional electric field and ion distribution model was developed using COMSOL to elucidate the distribution of potassium ions in different structures. COMSOL Multiphysics 6.1 was employed to describe the migration, deposition, electrochemical potential distribution, and current transport processes of ions using a quadratic current distribution approach. The model utilized spherical particles with a radius of 150 nm to simulate smooth surface nanoparticles. Additionally, pores were introduced inside the shell of hollow sphere to simulate porous structure. The electric field, current, and ion distribution during the migration of potassium ions in the two types of samples were simulated. The initial concentration of potassium ions in the electrolyte was maintained at 3.0 M. The initial concentration of H$^+$ was 0.1 M, corresponding to a pH value of 1. The model was considered a multi-physics coupling of the Nernst-Planck equation, which

accounts for the mass transport of all species:

$$\frac{\partial c_i}{\partial t} + \frac{\partial J_i}{\partial x} = R_i \tag{3}$$

Where $c_i$ is the concentration of species $i$ (1 for $K^+$, 2 for $H^+$, 3 for $OH^-$), $J_i$ is the flux, and $R_i$ is the reaction rate of species $i$. The flux was a sum of fluxes due to diffusion and migration:

$$J_i = -D_i \frac{\partial c_i}{\partial x} - \frac{z_i D_i}{RT} F c_i \frac{\partial \phi}{\partial x} \tag{4}$$

Where $D_i$, $z_i$ is the concentration and charge of species ($D_1 = 1.95 \times 10^{-9}$ $m^2\ s^{-1}$, $D_2 = 9.31 \times 10^{-9}$ $m^2\ s^{-1}$, $D_3 = 5.27 \times 10^{-9}$ $m^2\ s^{-1}$, $z_1 = +1$, $z_2 = +1$, $z_1 = -1$). F, R, and T represent the Faraday constant, gas constant, and absolute temperature (T = 298 K), respectively, and $\phi$ is the electrostatic potential.

The electric current was used to simulate the distribution of electric field, the formula for calculating the electric field as follows:

$$E = -\nabla V \tag{5}$$

And the dielectric model follows the rules:

$$D = \varepsilon_0 \varepsilon_r E \tag{6}$$

where $\varepsilon_0$ is the dielectric constant of vacuum, $\varepsilon_r$ is the dielectric constant of the materials.

The rate of formation of species i ($R_i$) was determined from the water reduction reaction and water dissociation equilibrium using the equations:

$$R_{H^+} = -k_{w2} c_{H^+} c_{OH^-} + k_{w1}$$

$$R_{OH^-} = -k_{w2} c_{H^+} C_{OH^-} + k_{w1} + r_{OH^-}$$

$$R_{H_2O} = -k_{w1} + k_{w2} c_{H^+} c_{OH^-}$$

Where $r_{OH^-}$ is the rate of $OH^-$ generation, $k_{w1}$ is the rate constant of the water reduction reaction ($2.4e-2\ mol\ m^{-3}\ s^{-1}$), and $k_{w2}$ is the reverse rate constant of water dissociation ($2.4e+6\ mol\ m^{-3}\ s^{-1}$). The rate of $OH^-$ generation was calculated based on the equation:

$$r_{OH^-} = \frac{j}{nF} \tag{7}$$

where $j$, $n$, and F is the applied current density, electron transfer number, Faradic constant, respectively.

### Density functional theory (DFT) calculations'
First-principles calculations based on density functional theory (DFT) were employed in our study[49,50]. The calculations were performed within the generalized gradient approximation (GGA) using the Perdew-Burke-Ernzerhof (PBE) formulation[51]. To describe the ionic cores and account for valence electrons, we utilized projected augmented wave (PAW) potentials[52]. A plane wave basis set with a kinetic energy cutoff of 450 eV was employed. To allow for partial occupancies of the Kohn–Sham orbitals, we utilized the Gaussian smearing method with a width of 0.05 eV. For geometry and lattice size optimizations, Brillouin zone integration was performed with a $1 \times 1 \times 1$ Gamma $k$-point sampling[53]. The self-consistent calculations applied a convergence energy threshold of $10^{-5}$ eV. During the optimization process, both the geometry and lattice constants were optimized by imposing a maximum stress of $0.02\ eV\ Å^{-1}$ on each atom. To eliminate

artificial interactions between periodic images, a 17 Å vacuum layer was typically added to the surfaces. The weak interactions were treated using the DFT + D3 method, which incorporates empirical corrections following Grimme's scheme[54,55]. In the case of magnetic systems, we employed the spin polarization method. The isosurface level for the charge density difference was set at $0.001\ eV\ Å^{-3}$. The Gibbs free energy for each elementary step was calculated as:

$$G = E_{elec} + E_{ZPE} - TS \tag{8}$$

where $E_{elec}$ is the electronic energy at 0 K calculated by DFT, $E_{ZPE}$ is the zero-point energy term, and T is the absolute temperature (here 298.15 K). To obtain the temperature and energy state, the constrained ab initio molecular dynamics (AIMD) has been used with slow-growth method. At the beginning of MD simulation, models have been heated up to 300 K by velocity scaling over 1.5 ps and then equilibrated at 300 K for 5 ps with a 2-fs time step.

## Data availability
The data that support the plots within this paper are available in the Source data file. Additional data available from authors upon request. Source data are provided with this paper.

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

## Acknowledgements

The work was supported by National Natural Science Foundation of China (22203099, 22002172, 22293015 and 22121002), National Key Research and Development Program of China (2020YFA0710203), Beijing Natural Science Foundation (J210020), Strategic Priority Research Program (A) of the Chinese Academy of Sciences (XDA0390400), CAS Project for Young Scientists in Basic Research

(YSBR-050), Science Foundation of China University of Petroleum, Beijing (2462023BJRC034), and Photon Science Center for Carbon Neutrality. The X-ray absorption spectroscopy measurements were performed at Beamline 1W2B and 1W1B at Beijing Synchrotron Radiation Facility (BSRF).

## Author contributions

J.Q.F., X.F.S. and B.X.H. proposed the project, designed the experiments, and wrote the manuscript; J.Q.F. performed the whole experiments; L.M.W., X.N.S., and L.B.Z. assisted in analysing the experimental data; S.H.J., X.D.M. and X.X.T. conducted a part of characterizations. X.C.K. and Q.G.Z. participated in discussions. X.F.S. and B.X.H. supervised the whole project.

## Competing interests

The authors declare no competing interests.
