## [Peer Review File · Nature Communications]

REVIEWER COMMENTS

Reviewer #1 (Remarks to the Author):

This research article of Han et al. reported a La-doped Cu hollow spherical catalyst with superior catalytic activity towards CO₂ electroreduction to C₂⁺ products in acidic electrolyte. The performances of this catalyst (Faradaic efficiency, partial current density and single pass carbon efficiency) are higher than or among the highest of most reported catalysts in similar conditions. This research also assembles multiple techniques of characterization and simulations. However, the interpretations of the spectra and simulations are not solid enough to rationalize the high performance of this catalyst. Especially, existing form of La is unclear. Metallic La atoms on the surface of the catalyst are unlikely to be formed through this synthetic procedure and are unlikely to exist in acidic aqueous solution. I think this article should not be accepted by Nature Communication unless the existing form of La can be clearly solved and the promotion effect of La can be reasonably explained. Below are detailed comments.

1. Line 50 "It was reported that the presence of K⁺ in acid electrolyte could shield the electric field around cathode and impede the diffusion of H⁺ to the catalyst surface, thereby inhibiting the HER." This is inaccurate. Supporting electrolyte like K⁺ shield the electric field and thus impede the 'migration' instead of 'diffusion' of H⁺.

2. The standard reduction potential of La³⁺/La is -2.52 V vs SHE. Therefore, it is unlikely that La³⁺ can be reduced during the synthesis of La-doped Cu₂O and the in-situ reduction of La-doped Cu₂O to La-doped Cu. The authors used La 3d XPS to characterize the oxidation state of La. In line 98, the authors claim "The high-resolution La 3d XPS spectra suggested that La was in a slightly oxidized chemical state in La-Cu HS and La-Cu SS (Figure S9)." However, the spectra in Figure S9 are quite similar to that of La-doped Cu₂O (Figure 5c). The valent state of La cannot be simply deduced from the binding energy of La 3d 5/2 peak. The binding energy of La₂O₃ (834~835 eV) is even lower than that of metal La (836 eV). The authors should compare the La 3d XPS spectrum of the catalyst with that of La₂O₃, metallic La and, if possible, La-Cu alloy. XANES and EXAFS of La are also possible techniques to reveal the existing form of La in the catalyst. I doubt that the existing form of La in the pre-catalyst or the catalyst during electrolysis is oxide or hydroxide of La(III) instead of being reduced.

3. Following last comment, is La species on the catalyst stable in acidic electrolyte? Both metallic La and oxide/hydroxide of La(III) can be dissolved in acidic solution. The high local pH during electrolysis may protect La(III) oxide/hydroxide from being dissolved, but dissolving may still occur at the beginning and the ending of the electrolysis. The authors should measure the loss of La after the electrolysis. And the element distribution after electrolysis should also be measured to check whether the La species on the surface of the catalyst was dissolved while that in the inner part was retained. These informations are important to understand how the La species promote the catalytic performance.

4. The authors conducted Nernst-Planck simulations on COMSOL. The governing equations, boundary conditions and values of coefficients should be provided to help readers to understand how the electrolysis process was simulated. In Nernst-Planck simulation, electroneutrality is kept in the whole domain. The structure of electric double layer cannot be reflected by Nernst-Planck simulation. Moreover, according to Gauss theorem, the charging state of the electrode (namely the catalyst here) is not considered. Therefore, the statement in Line 126 "It could be attributed to the attractive effect of the negative charge at the surface, and the diffusion of K⁺ into the channels" is not an accurate deduction of the Nernst-Planck simulation.

5. The authors should detect whether KHCO_3 precipitate was formed during the electrolysis by XRD and SEM after CO_2 reduction. The authors claim that more K was detected by XPS on La-Cu HS than on La-Cu SS after electrolysis. This might imply that more KHCO_3 precipitate was formed on La-Cu HS.
6. Line 168 "0.05 M H_2SO_4 aqueous solution containing 3 M KCl was used as catholyte ($\text{pH} < 1$)". 0.05M H_2SO_4 - 3M KCl was used as the electrolyte. The pH of this solution should be definitely higher than 1 because the second proton of H_2SO_4 is not a strong acid ($\text{pK}_a2 = 2$).
7. About the DFT simulation, since I think metallic La atom should not be the existing form of La on the surface of catalyst, the DFT simulation in the present manuscript cannot explain the promotion effect of La on CO_2 reduction. The authors should first identify the existing form of La on the catalyst during electrolysis and then conduct DFT simulation based on the detected structure (such as oxide or hydroxide cluster of La(III)).

Reviewer #2 (Remarks to the Author):

Feng et al. have presented a comprehensive approach that integrates microenvironment modulation and catalytic activity tuning to enhance CO_2RR -to- C_2+ generation in an acidic electrolyte. Their work has resulted in an impressive C_2+ Faradaic efficiency (FE) of 86.2%, with a CO_2 single-pass conversion efficiency (SPCE) reaching 52.8%. This study might provide guidance on promoting the activity and selectivity of CO_2RR toward the C_2+ products generation. I recommend that the authors address the following comments for further consideration before this manuscript can be accepted for publication in Nature Communications.

1. In Figure 3D, the highest FE for C_2+ was achieved with a La/Cu ratio of 0.3. However, the computational models only account for the substitution of one La atom for a Cu atom on the surface, which differs from the synthesized samples. It would be valuable to investigate whether a higher concentration of La in the models might lead to a distinct reaction mechanism and energy landscape.
2. It is suggested to take into account the electrolyte, especially H_2O , to incorporate the solvation, field, and cation effects. This is crucial as the configuration and energetics of adsorbed OCCO deviate from what has been reported in the existing literature.
3. Could a stronger CO adsorption leads to higher CO coverage, potentially inducing a different reaction mechanism and altering the energetics? This aspect should be explored further.
4. Computationally, in DFT calculations, it would be beneficial to compare the CO_2RR with the hydrogen evolution reaction (HER) to provide a broader perspective on the performance and selectivity of catalysts.
5. In the introduction section, the authors suggest that the porous structure enhances the concentration of surface OH^- and reactants, thus promoting catalytic performance. However, they introduce La-doped Cu hollow spheres (La-Cu HS) without a clear rationale for the choice of La dopants over other elements known to enhance CO adsorption and stabilize OCCO intermediates. The authors should justify the selection of La either through computational or experimental means. Additionally, does the choice of La in achieving the high C_2+ FE not only relate to the catalyst itself but also correlate with the porous structure in tuning the microenvironment?
6. Can Cu and La-Cu be distinguished in COMSOL multiphysics finite-element-based simulations? Further details on this aspect would help enhance the understanding of the simulations.

7. When introducing the statement "CO₂ single-pass conversion efficiency (SPCE) could reach 52.8%," it is advisable for the authors to include the experimental conditions to provide context and transparency for the readers.

Reviewer #3 (Remarks to the Author):

In this manuscript, the authors successfully achieved an efficient reduction of CO₂ to C₂⁺ under acidic condition by combining the microenvironment modulation via the porous channels structure and the intrinsic catalytic activity enhancement via La doping. It is very interesting that COMSOL Multiphysics finite-element-based simulations was adopted to investigate the contribution of porous channels structure to enriching K⁺ concentration and enhancing local pH around the catalytic sites, which were further verified by XPS spectra and in-situ surface-enhanced Raman spectroscopy. Density functional theory combined with in-situ attenuated total reflection-surface-enhanced IR absorption spectroscopy proved that the doping of La favored the *CO generation and decreased the energy barrier of C-C coupling reaction on Cu sites. The optimized La-Cu HS show a superior electrocatalytic CO₂-to-C₂⁺ performance under acid condition with C₂⁺ products Faradaic efficiency of 86.2%, partial current density of -775.8 mA cm⁻², CO₂ single-pass conversion efficiency 52.8%. The characterizations are well performed and in good agreement with the discussions. Overall, we recommend the acceptance of the manuscript by Nature Communications after addressing these following questions.

1. The coordinate state of La in the catalyst is not analyzed in detail. It is necessary to further analyze the chemical state of La in the catalyst, such as XAS measurement.
2. In Figure 1 G, the binding energy of Cu 2p_{3/2} peak of La-Cu HS and La-Cu SS shifted to lower binding energy comparing to that of Cu HS, which was attributed to the charge transfer between Cu and La. However, why is there no Cu-La bond observed in the XAS results of Cu?
3. In the section of "Electrocatalytic CO₂RR performance", La-Cu HS, La-Cu SS, and Cu HS were evaluated. How the relationship between the properties of the surface channel and the catalytic activity? Could the surface channel of the target catalysts be tunned? A set of Cu SS with various channels and insightful discussion is suggested to be added.
4. In Figure S18, a volcano-shaped relationship can be observed between C₂⁺ products FE and La doping content, it is necessary to explain the reasons for the change of La doping amount and products Faraday efficiency through calculation or related experiments.
5. One of the advantages to try CO₂RR under acidic conditions is to inhibit the formation of carbonate on the electrode surface, which can improve the stability of the electrolysis. In Figure 3H, is it reasonable to wash the electrode every 5 hours during the long-term stability test of the electrode?
6. How about the catalytic activity and stability of the developed electrocatalytic systems compared to other Cu-based catalysts?

Responses to the comments of the reviewers

Reviewer 1:

This research article of Han et al. reported a La-doped Cu hollow spherical catalyst with superior catalytic activity towards CO₂ electroreduction to C₂₊ products in acidic electrolyte. The performances of this catalyst (Faradaic efficiency, partial current density and single pass carbon efficiency) are higher than or among the highest of most reported catalysts in similar conditions. This research also assembles multiple techniques of characterization and simulations. However, the interpretations of the spectra and simulations are not solid enough to rationalize the high performance of this catalyst. Especially, existing form of La is unclear. Metallic La atoms on the surface of the catalyst are unlikely to be formed through this synthetic procedure and are unlikely to exist in acidic aqueous solution. I think this article should not be accepted by Nature Communication unless the existing form of La can be clearly solved and the promotion effect of La can be reasonably explained. Below are detailed comments.

Response: We thank the referee very much for the comment, which help and guide us to think about and polish our work greatly. We have tried our best to answer the questions from the reviewer. According to the reviewer's suggestion, we supplemented La XAFS and XPS data to confirm that La species exist as single atomic species. Meanwhile, we conducted EDS and ICP analyses on the catalyst and electrolyte after CO₂RR test, respectively, confirming the stable presence of La in the catalyst. Additionally, we employed both computational and experimental methods to investigate different lanthanide metal element dopant, and the results indicated that La doping is more favorable for the formation of C₂₊ products. We hope that we have addressed all of the questions satisfactorily. Many thanks for your time and efforts. We believe our manuscript meets the high standard of the journal, and also hope the reviewer could agree with us.

1. Line 50 "It was reported that the presence of K⁺ in acid electrolyte could shield the electric field around cathode and impede the diffusion of H⁺ to the catalyst surface, thereby inhibiting the HER." This is inaccurate. Supporting electrolyte like K⁺ shield the electric field and thus impede the 'migration' instead of 'diffusion' of H⁺.

Response: We thank the referee again for the comment. We agree with the reviewer. The movement of ions under the influence of an electric field should be described as "migration" rather than "diffusion." We have carefully reviewed the manuscript and made the necessary corrections.

2. The standard reduction potential of La³⁺/La is -2.52 V vs SHE. Therefore, it is unlikely that La³⁺ can be reduced during the synthesis of La-doped Cu₂O and the in-situ reduction of La-doped Cu₂O to La-doped Cu. The authors used La 3d XPS to characterize the oxidation state of La. In line 98, the authors claim "The high-resolution La 3d

XPS spectra suggested that La was in a slightly oxidized chemical state in La-Cu HS and La-Cu SS (Figure S9).¹ However, the spectra in Figure S9 are quite similar to that of La-doped Cu₂O (Figure 5c). The valent state of La cannot be simply deduced from the binding energy of La 3d 5/2 peak. The binding energy of La₂O₃ (834~835 eV) is even lower than that of metal La (836 eV). The authors should compare the La 3d XPS spectrum of the catalyst with that of La₂O₃, metallic La and, if possible, La-Cu alloy. XANES and EXAFS of La are also possible techniques to reveal the existing form of La in the catalyst. I doubt that the existing form of La in the pre-catalyst or the catalyst during electrolysis is oxide or hydroxide of La(III) instead of being reduced.

Response: We thank the referee again for the comment. The standard reduction potential of La³⁺/La is -2.52 V vs SHE. It is approximately -2.83 V when referenced to Ag/AgCl in electrocatalysis system. In the process of testing CO₂RR performance using chronopotentiometry, when a current of -0.9 A was applied, the displayed voltage was approximately -3.3 V vs Ag/AgCl. Therefore, La³⁺ could be reduced under our testing conditions.

The reviewer's perspective is correct, the valent state of La cannot be simply deduced from the binding energy of La 3d_{5/2} peak. However, the La 3d_{5/2} split can be used for characterization of La species. We have compared the La 3d_{5/2} peak of La-Cu HS and La-Cu SS with that of La₂O₃ and La(OH)₃ in Figure S9 of the revised supporting information. The binding energy of the La 3d_{5/2} peak for La-Cu HS and La-Cu SS is 835.0 eV, which is similar to that of La₂O₃ and La(OH)₃. However, the La 3d_{5/2} split in La-Cu HS and La-Cu SS is significantly smaller than that in La₂O₃ and La(OH)₃. Therefore, the La species in the as-prepared catalysts are in oxidized state, but it is noticeably distinct from that of La₂O₃ and La(OH)₃. We have discussed them in the revised manuscript as "The La 3d XPS spectra of La-Cu HS and La-Cu SS were displayed in Figure S9. The La 3d region has well separated spin-orbit components. The binding energy of the La 3d_{5/2} peak for La-Cu HS and La-Cu SS is 835.0 eV, which is similar to that of La₂O₃ and La(OH)₃. However, the La 3d_{5/2} split in La-Cu HS and La-Cu SS is significantly smaller than that in La₂O₃ and La(OH)₃. Therefore, the La species in the as-prepared catalysts are in oxidized state, but it is noticeably distinct from that of La₂O₃ and La(OH)₃.^{34,35}" Please see them in Pages 5-6 of the revised manuscript.

We have further analyzed the La species in the catalyst by XAS measurement. Owing to the limited La content in the catalysts, it is challenging to acquire effective La XAS data under operando conditions. Consequently, we conducted CO₂RR experiments under high current density conditions at the XAFS beamline. Subsequently, we immediately collected the La L₃-edge spectra of the catalyst on the gas diffusion electrode taken out from the flow cell, resulting in the acquisition of *ex-situ* La XAS data for both La-Cu HS and La-Cu SS. The La L₃-edge XANES spectra and FT EXAFS spectra were presented as Figure S12 in the revised manuscript. The La L₃-edge XANES spectra showed that the white line intensity of La-Cu HS and La-Cu SS was notably similar but inferior to that of La(OH)₃ and La₂O₃, indicating the similarity of La species within La-Cu HS and La-Cu SS and their distinction from La(OH)₃ and La₂O₃. In the FT EXAFS spectra of La-Cu HS and La-Cu SS, a prominent peak at approximately 2.0 Å was observed. It could be attributed to La-O scattering, which was analogous to that of La(OH)₃ or La₂O₃.

The presence of La-O coordination may be associated with the oxidation of single atom alloy catalysts during *ex-situ* tests (*Nat. Catal.* 2019, 2, 495; *Nat. Nanotechnol.* 2021, 16, 1386). Notably, the absence of a La-La contributing peak around 4.0 Å in La-Cu HS and La-Cu SS indicates that the La species exist as single atomic species. Furthermore, an observed peak at approximately 4.5 Å in the FT EXAFS spectra of La-Cu HS and La-Cu SS can be attributed to Cu-La scattering (*Nat. Commun.* 2023, 14, 3767). In conclusion, we have successfully synthesized atomic La doped Cu catalysts.

In the revised manuscript, we have discussed them by “Unfortunately, we failed to obtain effective La XAS signals under operando conditions, due to low La content in the catalysts. We can only determine the La species in La-Cu HS and La-Cu SS by immediately collecting XAS data after CO₂RR testing. The spectra of La(OH)₃ and La₂O₃ were collected as reference samples for comparison. As revealed by La L₃-edge XANES spectra (Figure S12), the white line intensity of La-Cu HS and La-Cu SS was similar but lower than that of La(OH)₃ and La₂O₃. It suggested that the La species in La-Cu HS and La-Cu SS was similar, but it was different from those in La(OH)₃ and La₂O₃. One main peak at around 2.0 Å was observed in the FT EXAFS spectra of La-Cu HS and La-Cu SS, which was attributed to La-O coordination. The La-O coordination might be due to the oxidation of the single atom alloy catalysts during *ex-situ* tests.^{36, 37} There is no La-La coordination peak at around 4.0 Å in La-Cu HS and La-Cu SS, confirming that the La species exist as single atomic species in La-Cu HS and La-Cu SS without long-range coordination to other La centers. Moreover, there is a peak at around 4.5 Å in La-Cu HS and La-Cu SS FT EXAFS spectra which could be assign to the Cu-La scattering.³⁸ Taken together, we have successfully synthesized atomic La-doped Cu catalysts.” Please see them in Pages 6-7 of the revised manuscript.

Additionally, we have described the process of collecting La L₃-edge spectra in the Method section as “The La L₃-edge spectra of the samples were collected by *ex-situ* method. The catalyst was loaded on GDE and conducted CO₂RR test at current density of -900 mA cm⁻² for 400 s in flow cell, then the GDE was taken out from flow cell and immediately proceed with the data collection of the La L₃-edge spectra using fluorescence mode.” Please see them in Page 19 of the revised manuscript.

3. Following last comment, is La species on the catalyst stable in acidic electrolyte? Both metallic La and oxide/hydroxide of La(III) can be dissolved in acidic solution. The high local pH during electrolysis may protect La(III) oxide/hydroxide from being dissolved, but dissolving may still occur at the beginning and the ending of the electrolysis. The authors should measure the loss of La after the electrolysis. And the element distribution after electrolysis should also be measured to check whether the La species on the surface of the catalyst was dissolved while that in the inner part was retained. These information are important to understand how the La species promote the catalytic performance.

Response: We thank the referee again for the comment. According to the comment, the EDS mappings of La-Cu

HS after CO₂RR were showed in Figure S25 of the revised supporting information. It indicated that La element still existed on the surface of La-Cu HS, and the La content was 1.18wt%, similar to the La content of La-Cu HS before CO₂RR. Meanwhile, we conducted ICP-OES analysis on the electrolyte after CO₂RR, and the result showed that the La content in the electrolyte can be neglected. Therefore, we can confirm that the stable presence of atomic La in the catalyst during CO₂RR process.

In the revised manuscript, we have discussed them by “Furthermore, the EDS mapping confirmed that La element still existed on the surface of La-Cu HS after CO₂RR (Figure S25), and the La content was 1.18wt%, similar to the La content of La-Cu HS before CO₂RR. The XPS confirmed that no notable changes were observed over element composition after electrolysis (Figure S26). The result of ICP-OES indicated that the La content in the electrolyte after CO₂RR can be neglected, confirming that atomic La species in the catalyst were stable during CO₂RR. Taken together, the La-Cu HS exhibited great promise for applications involving the conversion of CO₂ into C₂₊ products in acidic system.” Please see them in Pages 12-13 of the revised manuscript.

4. The authors conducted Nernst-Planck simulations on COMSOL. The governing equations, boundary conditions and values of coefficients should be provided to help readers to understand how the electrolysis process was simulated. In Nernst-Planck simulation, electroneutrality is kept in the whole domain. The structure of electric double layer cannot be reflected by Nernst-Planck simulation. Moreover, according to Gauss theorem, the charging state of the electrode (namely the catalyst here) is not considered. Therefore, the statement in Line 126 "It could be attributed to the attractive effect of the negative charge at the surface, and the diffusion of K⁺ into the channels" is not an accurate deduction of the Nernst-Planck simulation.

Response: We thank the referee again for the comment. According to the comment, the details of COMSOL Multiphysics simulations have been added in the Method section of the revised manuscript as “A three-dimensional electric field and ion distribution model was developed using COMSOL to elucidate the distribution of potassium ions in different structures. COMSOL Multiphysics 6.1 was employed to describe the migration, deposition, electrochemical potential distribution, and current transport processes of ions using a quadratic current distribution approach. The model utilized spherical particles with a radius of 150 nm to simulate smooth surface nanoparticles. Additionally, pores were introduced inside the shell of hollow sphere to simulate porous structure. The electric field, current, and ion distribution during the migration of potassium ions in the two types of samples were simulated. The initial concentration of potassium ions in the electrolyte was maintained at 3.0 M. The initial concentration of H⁺ was 0.1 M, corresponding to a pH value of 1. The model was considered a multi-physics coupling of the Nernst-Planck equation, which accounts for the mass transport of all species:

$$\frac{\partial c_i}{\partial t} + \frac{\partial J_i}{\partial x} = R_i \quad (1)$$

Where c_i is the concentration of species i (1 for K^+ , 2 for H^+ , 3 for OH^-), J_i is the flux, and R_i is the reaction rate of species i . The flux was a sum of fluxes due to diffusion and migration:

$$J_i = -D_i \frac{\partial c_i}{\partial x} - \frac{z_i D_i}{RT} F c_i \frac{\partial \phi}{\partial x} \quad (2)$$

Where D_i , z_i is the concentration and charge of species ($D_1 = 1.95 \times 10^{-9} \text{ m}^2 \text{ s}^{-1}$, $D_2 = 9.31 \times 10^{-9} \text{ m}^2 \text{ s}^{-1}$, $D_3 = 5.27 \times 10^{-9} \text{ m}^2 \text{ s}^{-1}$, $z_1 = +1$, $z_2 = +1$, $z_3 = -1$). F , R , and T represent the Faraday constant, gas constant, and absolute temperature ($T = 298 \text{ K}$), respectively, and ϕ is the electrostatic potential.

The electric current was used to simulate the distribution of electric field, the formula for calculating the electric field as follows:

$$E = -\nabla V \quad (3)$$

And the dielectric model follows the rules:

$$D = \varepsilon_0 \varepsilon_r E \quad (4)$$

where ε_0 is the dielectric constant of vacuum, ε_r is the dielectric constant of the materials.

The rate of formation of species i (R_i) was determined from the water reduction reaction and water dissociation equilibrium using the equations:

$$\begin{aligned} R_{H^+} &= -k_{w2} c_{H^+} c_{OH^-} + k_{w1} \\ R_{OH^-} &= -k_{w2} c_{H^+} c_{OH^-} + k_{w1} + r_{OH^-} \\ R_{H_2O} &= -k_{w1} + k_{w2} c_{H^+} c_{OH^-} \end{aligned}$$

Where r_{OH^-} is the rate of OH^- generation, k_{w1} is the rate constant of the water reduction reaction ($2.4 \times 10^{-2} \text{ mol m}^{-3} \text{ s}^{-1}$), and k_{w2} is the reverse rate constant of water dissociation ($2.4 \times 10^6 \text{ mol m}^{-3} \text{ s}^{-1}$). The rate of OH^- generation was calculated based on the equation:

$$r_{OH^-} = \frac{j}{nF} \quad (5)$$

where j , n , and F is the applied current density, electron transfer number, Faradic constant, respectively." Please see them in Pages 19-20 of the revised manuscript.

Furthermore, we also calculated the charge distribution over hollow sphere with porous channel structure in shell and solid sphere structure without channels by COMSOL under different current conditions and the results were presented as Figure S14 in the revised supporting information. It can be seen that a pronounced localized enrichment of charge on the surface of the hollow sphere. A significantly higher charge density was observed in the vicinity of the channels. The accumulation of negative charges near the channels could attract K^+ , facilitating the diffusion of K^+ into the channels. In the revised manuscript, we have discussed them by "Meanwhile, we also simulated the negative charge distribution over the hollow sphere with porous channel structure in shell and solid sphere structure without channels by COMSOL. Figure S14 showed a pronounced localized enrichment of charge

on the surface of the hollow sphere under all current conditions, with a significantly higher charge density was observed in the vicinity of the channels. The accumulation of negative charges near the channels could attract K^+ , facilitating the diffusion of K^+ into the channels.” Please see them in Page 8 of the revised manuscript.

5. The authors should detect whether $KHCO_3$ precipitate was formed during the electrolysis by XRD and SEM after CO_2 reduction. The authors claim that more K was detected by XPS on La-Cu HS than on La-Cu SS after electrolysis. This might imply that more $KHCO_3$ precipitate was formed on La-Cu HS.

Response: We thank the referee again for the comment. According to the comment, the XRD patterns of the electrode after CO_2 reduction was showed in Figure S15 of the revised supporting information. No peaks corresponding to $KHCO_3$ precipitate was found. Meanwhile, we also tested the pH value of the electrolyte after CO_2RR , which was 0.90. According to Henderson–Hasselbalch equations, in aqueous solution pH values < 4.3 , the majority of CO_2 exists in its undissociated form, namely CO_2 molecules, so $KHCO_3$ precipitate could not formed on the electrode within our electrolysis system.

In the revised manuscript, we have discussed them by “To further verify the ability of hollow sphere structure to enrich K^+ , we measured the K 2p XPS spectra over the GDE loaded with La-Cu HS and La-Cu SS after CO_2RR under the same conditions (Figure 2C). It is worth noting that no peaks corresponding to $KHCO_3$ precipitate was found in the XRD patterns of the electrode after CO_2 reduction (Figure S15), and the pH of the electrolyte after CO_2RR was 0.9, which could exclude the presence of $KHCO_3$ precipitate on the GDE surface. The XPS results showed a significant increase of K^+ content on the GDE loaded with La-Cu HS compared with that loaded with La-Cu SS, experimentally proving the conclusion of the simulation results. The elevated concentration of K^+ is conducive to inhibiting HER.” Please see them in Page 8 of the revised manuscript.

6. Line 168 "0.05 M H_2SO_4 aqueous solution containing 3 M KCl was used as catholyte (pH < 1)". 0.05 M H_2SO_4 -3M KCl was used as the electrolyte. The pH of this solution should be definitely higher than 1 because the second proton of H_2SO_4 is not a strong acid ($pK_{a2} = 2$).

Response: We thank the referee again for the comment. The reviewer is right, the pH value of 0.05 M H_2SO_4 aqueous solution is higher than 1 due to the second ionization of H_2SO_4 is incomplete. We have also measured the pH value of the as-prepared 0.05M H_2SO_4 aqueous solution using pH meter, and the result was 1.27. However, when preparing a 3 M KCl aqueous solution using 0.05 M H_2SO_4 aqueous solution, the resulting solution pH value was measured to be 0.87, which was similar to the value reported by literature (*Nat. Commun.* 2022, 13, 7596; *ACS Catal.* 2022, 12, 2357). The KCl was purchased from Alfa Aesar China Co., Ltd. and purity is 99.8%. Residual HCl may be present in KCl, leading to a decrease in pH value. In the revised manuscript, we have discussed them by “A flow cell equipped with GDE was used to test the CO_2RR performance of La-Cu HS, La-Cu SS, and Cu HS under different

current densities, and 0.05 M H₂SO₄ aqueous solution containing 3 M KCl was used as catholyte (pH = 0.87).²¹”

Please see them in Page 11 of the revised manuscript.

7. About the DFT simulation, since I think metallic La atom should not be the existing form of La on the surface of catalyst, the DFT simulation in the present manuscript cannot explain the promotion effect of La on CO₂ reduction. The authors should first identify the existing form of La on the catalyst during electrolysis and then conduct DFT simulation based on the detected structure (such as oxide or hydroxide cluster of La(III)).

Response: We thank the referee again for the comment. As discussed above, atomic La-doped Cu catalyst was obtained in this work, and the presented models were appropriate for the reaction pathway study in DFT simulation.

Reviewer 2: Feng et al. have presented a comprehensive approach that integrates microenvironment modulation and catalytic activity tuning to enhance CO₂RR-to-C₂₊ generation in an acidic electrolyte. Their work has resulted in an impressive C₂₊ Faradaic efficiency (FE) of 86.2%, with a CO₂ single-pass conversion efficiency (SPCE) reaching 52.8%. This study might provide guidance on promoting the activity and selectivity of CO₂RR toward the C₂₊ products generation. I recommend that the authors address the following comments for further consideration before this manuscript can be accepted for publication in Nature Communications.

Response: We thank the referee very much for the comment, which help and guide us to think about and polish our work greatly. We have tried our best to answer the questions from the reviewers. We hope that we have addressed all of the questions satisfactorily.

1. In Figure 3D, the highest FE for C₂₊ was achieved with a La/Cu ratio of 0.3. However, the computational models only account for the substitution of one La atom for a Cu atom on the surface, which differs from the synthesized samples. It would be valuable to investigate whether a higher concentration of La in the models might lead to a distinct reaction mechanism and energy landscape.

Response: We thank the referee again for the comment. According to the reviewer's suggestion, we have constructed the computational models that account for the substitution of 1, 2, 4, 6 La atoms for Cu atoms on the Cu(111) surface, respectively, and presented them in Figures S30-S34 of the revised supporting information. Subsequently, we calculated the Gibbs free energy change of the 2*CO-to-O*CCO step ($\Delta G_{(2^*CO_{-}to_{-}O^*CCO)}$) (Figure 4D). It can be seen that as the La atoms increased, the $\Delta G_{(2^*CO_{-}to_{-}O^*CCO)}$ initially decreased and then increased. When the number of doped La atoms was 4, the $\Delta G_{(2^*CO_{-}to_{-}O^*CCO)}$ exhibited the lowest value of 0.15 eV, which was much smaller than that of Cu(111) (1.39 eV), further demonstrating that the La dopants was favorable to C-C coupling process, and thus improving C₂₊ products selectivity.

In the revised manuscript, we have discussed them by “In order to investigate the effect of La atom content on the C-C coupling reaction, we replaced surface Cu atoms with 1, 2, 4, and 6 La atoms (denoted as 1La-Cu, 2La-Cu, 4La-Cu, 6La-Cu), constructed models with varying La atom contents, and calculated the Gibbs free energy change of the 2*CO-to-O*CCO step ($\Delta G_{(2^*CO_{-}to_{-}O^*CCO)}$). Notably, to account for the influence of H₂O and K⁺ on the reaction while simultaneously reducing computational complexity, we simplified the model by placing one K⁺ hydrated with six water molecules on the surface of the constructed model (Figures S30-S34).²¹ As shown in Figure 4D, with the increase in the number of La atoms doping, the $\Delta G_{(2^*CO_{-}to_{-}O^*CCO)}$ gradually decreased. When the number of doped La atoms was 4, the $\Delta G_{(2^*CO_{-}to_{-}O^*CCO)}$ reached its minimum value of 0.15 eV. Further increasing the number of doped La atoms resulted in the increase of $\Delta G_{(2^*CO_{-}to_{-}O^*CCO)}$. The results indicated that a moderate amount of La doping was favorable to C-C coupling process, which aligned with the volcano-shaped relationship observed in the experiments between C₂₊ products FE and La doping content.” Please see them in Page 15 of the revised manuscript.

2. It is suggested to take into account the electrolyte, especially H₂O, to incorporate the solvation, field, and cation effects. This is crucial as the configuration and energetics of adsorbed OCCO deviate from what has been reported in the existing literature.

Response: We thank the referee again for the comment. Considering the solvation effect can indeed bring the calculations closer to real reaction conditions. According to the comment, in order to incorporate the influence of H₂O molecules and K⁺ as much as possible in the calculation process, we modified the models by referencing methods described in the literature (*Nat. Commun.* 2022, 13, 7594), specifically by placing one K⁺ hydrated with six water molecules on the surface of the constructed model. All calculations related to free energy in the manuscript take into account the influence of H₂O molecules and K⁺. The adsorption forms of the reaction intermediates on the models were showed in Figures S30-36 of the revised supporting information. In the revised manuscript, we have discussed them by “Notably, to account for the influence of H₂O and K⁺ on the reaction while simultaneously reducing computational complexity, we simplified the model by placing one K⁺ hydrated with six water molecules on the surface of the constructed model.” Please see them in Page 15 of the revised manuscript.

3. Could a stronger CO adsorption leads to higher CO coverage, potentially inducing a different reaction mechanism and altering the energetics? This aspect should be explored further.

Response: We thank the referee again for the comment. A stronger CO adsorption would lead to higher CO coverage. It has been reported that within an appropriate range of CO coverage, as the CO coverage increases, the energy barrier for CO coupling gradually decreases, facilitating the occurrence of the C-C coupling reaction (*J. Am. Chem. Soc.* 2023, 145, 8714-8725; *ACS Catal.* 2017, 7, 1749-1756; *Surf. Sci.* 2016, 654, 56-62). According to the comment, we also employed DFT calculations to investigate the Gibbs free energy change of the 2*CO-to-O*CCO step on the surface of the 4La-Cu model under different *CO coverages. However, during the optimization of the adsorption structures of *CO at different coverages, we observed that when the coverage increased to a certain extent, CO spontaneously coupled, affecting our assessment of the coupling step. We had to individually place 2, 3, 5, and 7 *CO on the model surface to simulate changes in coverage. The results showed that as the number of *CO increases, the energy barrier for the C-C coupling step indeed gradually decreased, consistent with the reported trend in the literature.

4. Computationally, in DFT calculations, it would be beneficial to compare the CO₂RR with the hydrogen evolution reaction (HER) to provide a broader perspective on the performance and selectivity of catalysts.

Response: We thank the referee again for the comment. According to the comment, the hydrogen evolution reaction over Cu(111) and Cu(111) doped with 4 La atoms (4La-Cu(111)) were calculated considering the influence of H₂O

and K^+ , and the results were presented in the Figure S37 of the revised manuscript. The HER energy barrier over 4La-Cu(111) was 0.40 eV, which was larger than that over Cu(111) (0.31 eV), indicating that the doping of La is beneficial for suppressing HER reaction. Additionally, we adopted the difference between the thermodynamic limiting potentials (ΔU_L) for CO₂-to-O*CCO and HER (*i.e.*, $U_L(\text{CO}_2\text{RR})-U_L(\text{HER})$) to reflect the competing relationship between CO₂RR and HER (*Angew. Chem., Int. Ed.* 2018, 57, 12790; *J. Am. Chem. Soc.* 2017, 139, 8329). The more positive ΔU_L corresponds to higher selectivity toward CO₂ reduction. The potential limiting step of CO₂-to-*OCCO over Cu was 2*CO to O*CCO step, corresponding to $U_L(\text{CO}_2\text{RR})$ was -1.08 eV, while that over 4La-Cu was *CO₂ to *COOH step, corresponding to $U_L(\text{CO}_2\text{RR})$ was -0.74 eV. The ΔU_L of 4La-Cu(111) and Cu(111) were displayed in Figure 4F of the revised manuscript. The 4La-Cu(111) showed more positive ΔU_L than Cu, indicating that 4La-Cu was more prone to undergoing CO₂RR compared to Cu.

In the revised manuscript, we have discussed them by “Figure 4E displays the Gibbs free energy diagrams for CO₂ to *OCCO on 4La-Cu and Cu models (Figures S35, S36). The *CO₂ formation was an exothermic process on 4La-Cu, suggesting that the CO₂ molecule was favorable be adsorbed on 4La-Cu surface. The *OCCO formation from 2*CO was the potential limiting step for the Cu model. However, when 4 La atoms were doped onto the Cu surface, the $\Delta G_{(2*CO\text{-to-O*CCO})}$ decreased from 1.39 eV to 0.15 eV, further demonstrating that the La dopants was favorable to C-C coupling process, and thus improving C₂₊ products selectivity. The HER over Cu and 4La-Cu were calculated (Figure S37). The HER energy barrier over 4La-Cu was 0.40 eV, which was larger than that over Cu (0.31 eV), indicating that the doping of La is beneficial for suppressing HER reaction. Additionally, we adopted the difference between the thermodynamic limiting potentials (ΔU_L) for CO₂-to-*OCCO and HER (*i.e.*, $U_L(\text{CO}_2\text{RR})-U_L(\text{HER})$) to compare the selectivity of CO₂RR and HER,^{50, 51} and more positive ΔU_L corresponds to higher selectivity toward CO₂ reduction. The potential limiting step of CO₂-to-*OCCO over Cu was 2*CO to O*CCO step, corresponding to $U_L(\text{CO}_2\text{RR})$ was -1.08 eV, while that over 4La-Cu was *CO₂ to *COOH step, corresponding to $U_L(\text{CO}_2\text{RR})$ was -0.74 eV. The $U_L(\text{CO}_2\text{RR})$, $U_L(\text{HER})$ and ΔU_L of 4La-Cu and Cu were displayed in Figure 4F, the result suggested that 4La-Cu was more prone to undergoing CO₂RR compared to Cu.” Please see them in Pages 15-16 of the revised manuscript.

5. In the introduction section, the authors suggest that the porous structure enhances the concentration of surface OH⁻ and reactants, thus promoting catalytic performance. However, they introduce La-doped Cu hollow spheres (La-Cu HS) without a clear rationale for the choice of La dopants over other elements known to enhance CO adsorption and stabilize OCCO intermediates. The authors should justify the selection of La either through computational or experimental means. Additionally, does the choice of La in achieving the high C₂₊ FE not only relate to the catalyst itself but also correlate with the porous structure in tuning the microenvironment?

Response: We thank the referee again for the comment, and we answer the two questions as follows.

1) Doping another metal element into Cu-based catalyst has been demonstrated as an effective method for optimizing the binding strength of intermediates, and enhancing the intrinsic activity of electrocatalytic CO₂-to-C₂₊ products. Different from d-block metal elements, lanthanide metal (LM) elements possess intense spin-orbit coupling and lanthanide contraction effects. These properties lead to the accumulation of localized electronic states and have the potential to alter the electronic structure of doped d-block metal species, thereby enabling catalytic enhancement. Hence, we employed both computational and experimental methods to select appropriate dopant elements from lanthanide metal element. We screened four lanthanide metal dopants (La, Pr, Tb, Er) dispersed in Cu(111) lattice (La-Cu, Pr-Cu, Tb-Cu, Er-Cu) and calculated the Gibbs free energy change of the 2*CO-to-O*CCO step ($\Delta G_{(2*CO-to-O*CCO)}$) to assess their activity in CO₂RR to C₂₊ products, taking into account the influence of H₂O and potassium. The constructed models and calculation results were showed in Figures S38-S42 of the revised supporting information. The $\Delta G_{(2*CO-to-O*CCO)}$ of La-Cu was significantly lower than that of Pr-Cu, Tb-Cu and Er-Cu, suggesting that La doping is more effective in promoting C-C coupling. Meanwhile, we synthesized Pr-Cu, Tb-Cu, and Er-Cu using the same method as La-Cu and tested their CO₂RR activity under identical electrolyte and current density conditions. The products distribution was presented in Figure S43 of revised supporting information, and the results indicated that La-Cu exhibited highest C₂₊ products FE, consistent with the DFT calculations. Both computational and experimental results confirmed that La doping is more favorable for the formation of C₂₊ products.

In the revised manuscript, we have discussed them by “Additionally, enhancing the intrinsic activity of Cu-based catalyst for CO₂RR is also vital. Doping another metal element into Cu-based catalyst has been demonstrated as an effective method for tuning the binding strength of intermediates and thus enhancing the intrinsic activity of electrocatalytic CO₂-to-C₂₊ products.^{28, 29} Different from d-block metal elements, lanthanide metal (LM) elements possess intense spin-orbit coupling and lanthanide contraction effects. These properties lead to the accumulation of localized electronic states and hold the potential to alter the electronic structure of doped d-block metal species, thus enabling catalytic enhancement.^{30, 31} Therefore, it can be anticipated that designing a LM-doped Cu-based catalyst with the porous channel structure to enrich K⁺ and form a high alkaline microenvironment would significantly enhance the performance of CO₂RR to C₂₊ products in an acidic system.” Please see them in Page 4 of the revised manuscript.

“In addition to La, we also investigated the influence of doping other LM elements (Pr, Tb, and Er) into Cu on CO₂RR to C₂₊ products. The models of 4Pr-Cu, 4Tb-Cu and 4Er-Cu were constructed and the calculated $\Delta G_{(2*CO-to-O*CCO)}$ were displayed in Figures S38-S42. Among 4Pr-Cu, 4Tb-Cu, 4Er-Cu and 4La-Cu, the 4La-Cu displayed a lower $\Delta G_{(2*CO-to-O*CCO)}$ than that of Pr-Cu, Tb-Cu and Er-Cu, suggesting that La doping is more effective in promoting C-C coupling. Meanwhile, we synthesized Pr-Cu, Tb-Cu, and Er-Cu catalysts using the same method as La-Cu HS and tested their CO₂RR activity under identical electrolyte and current density conditions. The products distribution was presented in Figure S43, and the results indicated that La-Cu exhibited highest C₂₊ products FE,

consistent with the DFT calculations. Both computational and experimental results confirmed that La doping is more favorable for the formation of C₂₊ products.” Please see them in Page 16 of the revised manuscript.

2) The reviewer is right, the choice of La in achieving the high C₂₊ FE not only relate to the catalyst itself but also correlate with the porous structure to tune the microenvironment. Experimental and theoretical studies indicated that the La doping facilitated C-C coupling. On the other hand, the channel structure played a crucial role in accumulating K⁺ and OH⁻ species near the catalyst surface and within the channels, which effectively suppressed the undesired HER and promoted the C-C coupling. These combined effects significantly promote the activity and selectivity of CO₂RR toward the production of C₂₊ products.

6. Can Cu and La-Cu be distinguished in COMSOL Multiphysics finite-element-based simulations? Further details on this aspect would help enhance the understanding of the simulations.

Response: We thank the referee again for the comment. COMSOL Multiphysics is typically used for macro-scale engineering and scientific simulations, primarily focusing on macro-scale physical phenomena such as electromagnetic fields, heat conduction, fluid flow, structural stress, and more, which cannot distinguish Cu and Cu-La. Therefore, we employed DFT calculations in the later sections of the manuscript to discuss the impact of La doping on the CO₂RR activity. The details of COMSOL Multiphysics simulations have been added in the Method section of the revised manuscript as “A three-dimensional electric field and ion distribution model was developed using COMSOL to elucidate the distribution of potassium ions in different structures. COMSOL Multiphysics 6.1 was employed to describe the migration, deposition, electrochemical potential distribution, and current transport processes of ions using a quadratic current distribution approach. The model utilized spherical particles with a radius of 150 nm to simulate smooth surface nanoparticles. Additionally, pores were introduced inside the shell of hollow sphere to simulate porous structure. The electric field, current, and ion distribution during the migration of potassium ions in the two types of samples were simulated. The initial concentration of potassium ions in the electrolyte was maintained at 3.0 M. The initial concentration of H⁺ was 0.1 M, corresponding to a pH value of 1. The model was considered a multi-physics coupling of the Nernst-Planck equation, which accounts for the mass transport of all species:

$$\frac{\partial c_i}{\partial t} + \frac{\partial J_i}{\partial x} = R_i \quad (6)$$

Where c_i is the concentration of species i (1 for K⁺, 2 for H⁺, 3 for OH⁻), J_i is the flux, and R_i is the reaction rate of species i . The flux was a sum of fluxes due to diffusion and migration:

$$J_i = -D_i \frac{\partial c_i}{\partial x} - \frac{z_i D_i}{RT} F c_i \frac{\partial \phi}{\partial x} \quad (7)$$

Where D_i , z_i is the concentration and charge of species ($D_1 = 1.95 \times 10^{-9} \text{ m}^2 \text{ s}^{-1}$, $D_2 = 9.31 \times 10^{-9} \text{ m}^2 \text{ s}^{-1}$, $D_3 = 5.27 \times 10^{-9} \text{ m}^2 \text{ s}^{-1}$, $z_1 = +1$, $z_2 = +1$, $z_3 = -1$). F, R, and T represent the Faraday constant, gas constant, and absolute

temperature ($T = 298 \text{ K}$), respectively, and ϕ is the electrostatic potential.

The electric current was used to simulate the distribution of electric field, the formula for calculating the electric field as follows:

$$E = -\nabla V \quad (8)$$

And the dielectric model follows the rules:

$$D = \varepsilon_0 \varepsilon_r E \quad (9)$$

where ε_0 is the dielectric constant of vacuum, ε_r is the dielectric constant of the materials.

The rate of formation of species i (R_i) was determined from the water reduction reaction and water dissociation equilibrium using the equations:

$$R_{H^+} = -k_{w2} c_{H^+} c_{OH^-} + k_{w1}$$

$$R_{OH^-} = -k_{w2} c_{H^+} c_{OH^-} + k_{w1} + r_{OH^-}$$

$$R_{H_2O} = -k_{w1} + k_{w2} c_{H^+} c_{OH^-}$$

Where r_{OH^-} is the rate of OH^- generation, k_{w1} is the rate constant of the water reduction reaction ($2.4 \times 10^{-2} \text{ mol m}^{-3} \text{ s}^{-1}$), and k_{w2} is the reverse rate constant of water dissociation ($2.4 \times 10^6 \text{ mol m}^{-3} \text{ s}^{-1}$). The rate of OH^- generation was calculated based on the equation:

$$r_{OH^-} = \frac{j}{nF} \quad (10)$$

where j , n , and F is the applied current density, electron transfer number, Faradic constant, respectively." Please see them in Pages 19-20 of the revised manuscript.

7. When introducing the statement "CO₂ single-pass conversion efficiency (SPCE) could reach 52.8%," it is advisable for the authors to include the experimental conditions to provide context and transparency for the readers.

Response: We thank the referee again for the comment. We have modified the statement as "CO₂ single-pass conversion efficiency (SPCE) for C₂₊ products could reach 52.8% at -900 mA cm^{-2} , when the CO₂ flow rate was 3 standard cubic centimeters per minute." Please see them in Page 2 of the revised manuscript.

Reviewer 3: In this manuscript, the authors successfully achieved an efficient reduction of CO₂ to C₂₊ under acidic condition by combining the microenvironment modulation via the porous channels structure and the intrinsic catalytic activity enhancement via La doping. It is very interesting that COMSOL Multiphysics finite-element-based simulations was adopted to investigate the contribution of porous channels structure to enriching K⁺ concentration and enhancing local pH around the catalytic sites, which were further verified by XPS spectra and in-situ surface-enhanced Raman spectroscopy. Density functional theory combined with in-situ attenuated total reflection-surface-enhanced IR absorption spectroscopy proved that the doping of La favored the *CO generation and decreased the energy barrier of C-C coupling reaction on Cu sites. The optimized La-Cu HS show a superior electrocatalytic CO₂-to-C₂₊ performance under acid condition with C₂₊ products Faradaic efficiency of 86.2%, partial current density of -775.8 mA cm⁻², CO₂ single-pass conversion efficiency 52.8%. The characterizations are well performed and in good agreement with the discussions. Overall, we recommend the acceptance of the manuscript by Nature Communications after addressing these following questions.

Response: We thank the referee very much for the comment, which help and guide us to think about and polish our work greatly. We have tried our best to answer the questions from the reviewers. We hope that we have addressed all of the questions satisfactorily.

1. The coordinate state of La in the catalyst is not analyzed in detail. It is necessary to further analyze the chemical state of La in the catalyst, such as XAS measurement.

Response: We thank the referee again for the comment. According to comment, we have further analyzed the La species in the catalyst by XAS measurement. Owing to the limited La content in the catalysts, it is challenging to acquire effective La XAS data under operando conditions. Consequently, we conducted CO₂RR experiments under high current density conditions at the XAFS beamline. Subsequently, we immediately collected the La L₃-edge spectra of the catalyst on the gas diffusion electrode taken out from the flow cell, resulting in the acquisition of *ex-situ* La XAS data for both La-Cu HS and La-Cu SS. The La L₃-edge XANES spectra and FT EXAFS spectra were presented as Figure S12 in the revised manuscript. The La L₃-edge XANES spectra showed that the white line intensity of La-Cu HS and La-Cu SS was notably similar but inferior to that of La(OH)₃ and La₂O₃, indicating the similarity of La species within La-Cu HS and La-Cu SS and their distinction from La(OH)₃ and La₂O₃. In the FT EXAFS spectra of La-Cu HS and La-Cu SS, a prominent peak at approximately 2.0 Å was observed. It could be attributed to La-O scattering, which was analogous to that of La(OH)₃ or La₂O₃. The presence of La-O coordination may be associated with the oxidation of single atom alloy catalysts during *ex-situ* tests (*Nat. Catal.* 2019, 2, 495; *Nat. Nanotechnol.* 2021, 16, 1386). Notably, the absence of a La-La contributing peak around 4.0 Å in La-Cu HS and La-Cu SS confirms that the La species exist as single atomic species. Furthermore, an observed peak at approximately 4.5 Å in the FT EXAFS spectra of La-Cu HS and La-Cu SS can be attributed to Cu-La scattering

(*Nat. Commun.* 2023, 14, 3767). In conclusion, we have successfully synthesized atomic La doped Cu catalysts.

In the revised manuscript, we have discussed them by “Unfortunately, we failed to obtain effective La XAS signals under operando conditions, due to low La content in the catalysts. We can only determine the La species in La-Cu HS and La-Cu SS by immediately collecting XAS data after CO₂RR testing. The spectra of La(OH)₃ and La₂O₃ were collected as reference samples for comparison. As revealed by La L₃-edge XANES spectra (Figure S12), the white line intensity of La-Cu HS and La-Cu SS was similar but lower than that of La(OH)₃ and La₂O₃. It suggested that the La species in La-Cu HS and La-Cu SS was similar, but it was different from those in La(OH)₃ and La₂O₃. One main peak at around 2.0 Å was observed in the FT EXAFS spectra of La-Cu HS and La-Cu SS, which was attributed to La-O coordination. The La-O coordination might be due to the oxidation of the single atom alloy catalysts during *ex-situ* tests.^{36, 37} There is no La-La coordination peak at around 4.0 Å in La-Cu HS and La-Cu SS, confirming that the La species exist as single atomic species in La-Cu HS and La-Cu SS without long-range coordination to other La centers. Moreover, there is a peak at around 4.5 Å in La-Cu HS and La-Cu SS FT EXAFS spectra which could be assign to the Cu-La scattering.³⁸ Taken together, we have successfully synthesized atomic La-doped Cu catalysts.” Please see them in Pages 6-7 of the revised manuscript.

2. In Figure 1 G, the binding energy of Cu 2p_{3/2} peak of La-Cu HS and La-Cu SS shifted to lower binding energy comparing to that of Cu HS, which was attributed to the charge transfer between Cu and La. However, why is there no Cu-La bond observed in the XAS results of Cu?

Response: We thank the referee again for the comment. The absence of a Cu-La contributing peak in the Cu FT EXAFS spectra may potentially result from the strong signal of Cu, which could mask the Cu-La peak. According to comment, we have further analyzed the La species in the catalyst by XAS measurement, the results were displayed in Figure S12 of the revised manuscript. In the La L₃-edge FT EXAFS spectra of La-Cu HS and La-Cu SS, a prominent peak at approximately 2.0 Å was observed. It could be attributed to La-O scattering, which was analogous to that of La(OH)₃ or La₂O₃. The presence of La-O coordination may be associated with the oxidation of single atom alloy catalysts during *ex-situ* tests (*Nat. Catal.* 2019, 2, 495; *Nat. Nanotechnol.* 2021, 16, 1386). Notably, the absence of a La-La contributing peak around 4.0 Å in La-Cu HS and La-Cu SS confirms that the La species exist as single atomic species. Furthermore, an observed peak at approximately 4.5 Å in the FT EXAFS spectra of La-Cu HS and La-Cu SS can be attributed to Cu-La scattering (*Nat. Commun.* 2023, 14, 3767). In conclusion, we have successfully synthesized atomic La doped Cu catalysts.

In the revised manuscript, we have discussed them by “One main peak at around 2.0 Å was observed in the FT EXAFS spectra of La-Cu HS and La-Cu SS, which was attributed to La-O coordination. The La-O coordination might be due to the oxidation of the single atom alloy catalysts during *ex-situ* tests.^{36, 37} There is no La-La coordination peak at around 4.0 Å in La-Cu HS and La-Cu SS, confirming that the La species exist as single atomic

species in La-Cu HS and La-Cu SS without long-range coordination to other La centers. Moreover, there is a peak at around 4.5 Å in La-Cu HS and La-Cu SS FT EXAFS spectra which could be assigned to the Cu-La scattering.³⁸ Taken together, we have successfully synthesized atomic La-doped Cu catalysts.” Please see them in Pages 6-7 of the revised manuscript.

3. In the section of “Electrocatalytic CO₂RR performance”, La-Cu HS, La-Cu SS, and Cu HS were evaluated. How the relationship between the properties of the surface channel and the catalytic activity? Could the surface channel of the target catalysts be tuned? A set of Cu SS with various channels and insightful discussion is suggested to be added.

Response: We are very grateful to the reviewer for the comment, which involves investigating the effect of pore properties on CO₂RR activity by manipulating the catalyst surface porosity. However, the La-Cu HS in this work was synthesized through inside-out Ostwald ripening mechanism (*Nano. Res.* 2016, 9, 1891; *Adv. Mater.* 2006, 18, 2325), which includes three stages: (1) precipitation of solid cores, (2) subsequent deposition of the outer shell on the surface of cores, and (3) core dissolution and shell recrystallization. Due to the absence of template agents in the catalyst synthesis process, it is hard to achieve precise control over the surface channel structure properties of the catalyst. We think this is an interesting research topic.

4. In Figure S18, a volcano-shaped relationship can be observed between C₂₊ products FE and La doping content, it is necessary to explain the reasons for the change of La doping amount and products Faraday efficiency through calculation or related experiments.

Response: We thank the referee again for the comment. According to the comment, we constructed the computational models that account for the substitution of 1, 2, 4, 6 La atoms for Cu atoms on the surface, respectively, and presented them in Figures S30-S34 of the revised supporting information. Subsequently, the Gibbs free energy change of the 2*CO-to-O*CCO step ($\Delta G_{(2*CO-to-O*CCO)}$) were calculated, and the results were presented in Figure 4D of the revised manuscript. It can be seen that with the increase in the number of La atoms doping, the $\Delta G_{(2*CO-to-O*CCO)}$ gradually decreased. When the number of doped La atoms was 4, the $\Delta G_{(2*CO-to-O*CCO)}$ reached its minimum value. Further increasing the number of doped La atoms resulted in the increase of $\Delta G_{(2*CO-to-O*CCO)}$. The above results indicate that only a moderate amount of La doping enhances the selectivity of C₂₊ products. This aligns with the volcano-shaped relationship observed in the experiments between C₂₊ products FE and La doping content.

In the revised manuscript, we have discussed them by “In order to investigate the effect of La atom content on the C-C coupling reaction, we replaced surface Cu atoms with 1, 2, 4, and 6 La atoms (denoted as 1La-Cu, 2La-Cu, 4La-Cu, 6La-Cu), constructed models with varying La atom contents, and calculated the Gibbs free energy change of the 2*CO-to-O*CCO step ($\Delta G_{(2*CO-to-O*CCO)}$). Notably, to account for the influence of H₂O and K⁺ on the reaction

while simultaneously reducing computational complexity, we simplified the model by placing one K^+ hydrated with six water molecules on the surface of the constructed model (Figures S30-S34).²¹ As shown in Figure 4D, with the increase in the number of La atoms doping, the $\Delta G_{(2*CO_{10}-O*CCO)}$ gradually decreased. When the number of doped La atoms was 4, the $\Delta G_{(2*CO_{10}-O*CCO)}$ reached its minimum value of 0.15 eV. Further increasing the number of doped La atoms resulted in the increase of $\Delta G_{(2*CO_{10}-O*CCO)}$. The results indicated that a moderate amount of La doping was favorable to C-C coupling process, which aligned with the volcano-shaped relationship observed in the experiments between C_{2+} products FE and La doping content.” Please see them in Page 15 of the revised manuscript.

5. One of the advantages to try CO_2RR under acidic conditions is to inhibit the formation of carbonate on the electrode surface, which can improve the stability of the electrolysis. In Figure 3H, is it reasonable to wash the electrode every 5 hours during the long-term stability test of the electrode?

Response: We thank the referee again for the comment. The CO_2RR stability is related with catalyst, electrolyte, gas diffusion electrode (GDE) and so on. In flow cell, CO_2RR occurred at the three-phase interface formed by GDE with CO_2 gas channel and hydrophobic layer. CO_2 can react with hydroxide in electrolyte to form carbonate precipitation during CO_2RR process, and the accumulation of carbonate at three-phase interface would block CO_2 gas channel and the gas diffusion electrode suffers from losing hydrophobicity and being flooded over time. Both CO_2 gas channel blockage and flooding would destroy GDE stability, and thus affect the catalytic performance. Although the use of an acidic electrolyte system effectively addresses the issue of carbonate precipitation blocking CO_2 gas channels, the issue of gradual loss of hydrophobicity of GDE still remains, especially under high current density conditions. To alleviate the problem of the gradual loss of hydrophobicity of GDE, the CO_2RR was interrupted every 5 hours and the GDE were removed, washed with deionized water and followed by dryness under N_2 atmosphere.

In the revised manuscript, we have discussed them by “We also evaluated the long-term stability of the La-Cu HS in acidic system at -900 mA cm^{-2} . To alleviate the problem of the gradual loss of hydrophobicity of GDE, the CO_2RR was interrupted every 5 hours and the GDE were removed, washed with deionized water and followed by dryness under N_2 atmosphere.” Please see them in Page 12 of the revised manuscript.

6. How about the catalytic activity and stability of the developed electrocatalytic systems compared to other Cu-based catalysts?

Response: We thank the referee again for the comment. According to the comment, we have compared the catalytic activity and stability of the developed electrocatalytic systems in Table S1 of the revised supporting information. We can find that the as-synthesized La-Cu catalyst not only exhibited significant advantages in C_{2+} products selectivity and partial current density, but also possessed outstanding stability. Please see them in Table S1 of the

revised supporting information.

REVIEWER COMMENTS

Reviewer #1 (Remarks to the Author):

The revised manuscript has concerned all my comments and I really appreciate the efforts done by the authors. However, I am still not convinced about the existing form of La element in the catalyst. The proposed structure of the catalyst is like a La-Cu single-atom alloy or a La adatom on Cu surface (like Figure S28). The authors provided three major evidences.

1. The first evidence is the electrode potential. The authors claimed that the electrode potential at -0.9 A was more negative than the standard reduction potential of La^{3+}/La (-2.52 V vs SHE). However, according to Figure 3H, the electrode potential at -0.9 A was about -2 V vs RHE. Considering that the pH of electrolyte was about 1, the electrode potential was about -2.05 V vs SHE, more positive than the standard reduction potential of La^{3+}/La . Moreover, standard reduction potential needs the concentration of La^{3+} to be 1 M. To reduce La^{3+} species in the real condition needs much more negative potential.

2. La 3d XPS. "However, the La 3d_{5/2} split in La-Cu HS and La-Cu SS is significantly smaller than that in La_2O_3 and $\text{La}(\text{OH})_3$. Therefore, the La species in the as-prepared catalysts are in oxidized state, but it is noticeably distinct from that of La_2O_3 and $\text{La}(\text{OH})_3$." In Figure S9, I did not observe significant difference between the spectra of La-Cu catalysts and La_2O_3 or $\text{La}(\text{OH})_3$. What is the smaller split?

3. La L3-edge XAS. In the EXAFS curves, the authors claimed the absence of La-La path at 4.0 Å and the presence of La-Cu path at 4.5 Å. This length is more likely a La-O-Cu path. If La and Cu atoms form a bond like the cases in La-Cu alloy or La adatom on Cu surface, the La-Cu bond length should be shorter than 4 Å according to the atomic radii.

Therefore, based on the comments above, in my opinion, a more reasonable structure of the La-Cu catalysts should be La-Ox sites on Cu surface. These sites may be discrete. Thus, their properties may differ from La_2O_3 and $\text{La}(\text{OH})_3$. But I think metallic La atom sites are unlikely to be formed on Cu. The La-Ox site may also act as a Lewis acid site which can activate CO_2 and promote the CO_2 reduction process. I suggest the authors to conduct DFT simulations based on the model with La-Ox site on Cu surface to check whether this structure favors CO_2 reduction to C_2^+ products.

Reviewer #2 (Remarks to the Author):

The revised version of manuscript NCOMMS-23-44546A incorporates more rigorous DFT calculations, specifically examining solvation effects and doping concentration. However, I am hesitant to fully recommend its publication in Nature Communications unless the authors address the following concerns adequately:

1. I appreciate the authors' consideration of including water and cation in the revised models. However, I question the robustness of the DFT calculations due to the insufficient number of water molecules. It appears that, in some cases, $^*\text{OCCO}$ is stabilized by the interaction of hydrogen bonding and K^+-^*OCCO , but the K^+ ion is sometimes distant from the adsorbed $^*\text{CO}$.

2. The water structure seems distant from the real electrolyte; hence, the authors should consider

incorporating more water molecules to establish a continuous hydrogen network.

3. The authors need to clarify the principles used to determine these structures, whether through random sampling or identifying the most thermodynamically stable configuration. Conducting ab initio molecular dynamics simulations for an extended time might help ascertain whether these structures are trapped in local minima.

4. It is crucial for the authors to consider the corresponding initial and final states of 2^*CO and *OCCO . The identified 2^*CO structures under different concentrations might not be the most thermodynamically favored ones.

Addressing these concerns will bring more scientific rigour to this manuscript and enhance its suitability for publication in Nature Communications.

Reviewer #3 (Remarks to the Author):

All my concerns have been addressed. We recommend the acceptance of the manuscript by Nature Communications.

Responses to the comments of the reviewers

Reviewer 1:

The revised manuscript has concerned all my comments and I really appreciate the efforts done by the authors. However, I am still not convinced about the existing form of La element in the catalyst. The proposed structure of the catalyst is like a La-Cu single-atom alloy or a La adatom on Cu surface (like Figure S28). The authors provided three major evidences.

Response: We thank the referee very much for the comment, and we have carefully revised the manuscript based on the comments.

1. The first evidence is the electrode potential. The authors claimed that the electrode potential at -0.9 A was more negative than the standard reduction potential of La^{3+}/La (-2.52 V vs SHE). However, according to Figure 3H, the electrode potential at -0.9 A was about -2 V vs RHE. Considering that the pH of electrolyte was about 1, the electrode potential was about -2.05 V vs SHE, more positive than the standard reduction potential of La^{3+}/La . Moreover, standard reduction potential needs the concentration of La^{3+} to be 1 M. To reduce La^{3+} species in the real condition needs much more negative potential.

Response: We thank the referee again for the comment. According to the comments, we conducted a thorough analysis, and we agree with the referee. La^{3+} cannot be reduced to La^0 under the experimental conditions, and the presence of La should be in the form La-O_x sites on the Cu surface. XPS and XAS data provided sufficient evidence. In the revised manuscript, we have modified the discussion by “The La $3d_{5/2}$ XPS spectra of La-Cu HS and La-Cu SS were displayed in Figure S9, the La $3d_{5/2}$ region has well separated spin-orbit components. The binding energy of the La $3d_{5/2}$ peak for La-Cu HS and La-Cu SS is 835.0 eV, indicating that the La species is in oxidized state. Additionally, the La $3d_{5/2}$ split, denoted as ΔE , was marked in Figure S9. It can be observed that La-Cu HS and La-Cu SS show the similar ΔE , which is smaller than that of La_2O_3 and $\text{La}(\text{OH})_3$, suggesting that the La species oxidized state of La-Cu HS and La-Cu SS is distinct from that of La_2O_3 and $\text{La}(\text{OH})_3$.^{34, 35}” and “One main peak at around 2.0 Å was observed in the FT EXAFS spectra of La-Cu HS and La-Cu SS, which was attributed to La-O coordination. There is no La-La coordination peak at around 4.0 Å in La-Cu HS and La-Cu SS, confirming that the La species exists as single atomic species in La-Cu HS and La-Cu SS without long-range coordination to other La centers. Moreover, there is a peak at around 4.5 Å in La-Cu HS and La-Cu SS FT EXAFS spectra, which could be assigned to the La-O-Cu scattering.³⁶ Therefore, the La species in La-Cu HS and La-Cu SS are the form of La-O_x sites on the Cu surface.” Please see them in pages 5-7 of the revised manuscript.

2. La 3d XPS. "However, the La 3d_{5/2} split in La-Cu HS and La-Cu SS is significantly smaller than that in La₂O₃ and La(OH)₃. Therefore, the La species in the as-prepared catalysts are in oxidized state, but it is noticeably distinct from that of La₂O₃ and La(OH)₃." In Figure S9, I did not observe significant difference between the spectra of La-Cu catalysts and La₂O₃ or La(OH)₃. What is the smaller split?

Response: We thank the referee again for the comment. To better show the difference between the spectra of La-Cu catalysts and La₂O₃ or La(OH)₃, we magnified the 3d_{5/2} peak of La and presented it in Figure S9 in the revised supporting information. The La 3d_{5/2} split, denoted as ΔE , was marked in the Figure S9. It can be observed that La-Cu HS and La-Cu SS show the similar ΔE , which is smaller than that of La₂O₃ and La(OH)₃. Therefore, the La species in the as-prepared catalysts are in oxidized state, but it is noticeably distinct from that of La₂O₃ and La(OH)₃.

In the revised manuscript, we have discussed them by "The La 3d_{5/2} XPS spectra of La-Cu HS and La-Cu SS were displayed in Figure S9, the La 3d_{5/2} region has well separated spin-orbit components. The binding energy of the La 3d_{5/2} peak for La-Cu HS and La-Cu SS is 835.0 eV, indicating that the La species is in oxidized state. Additionally, the La 3d_{5/2} split, denoted as ΔE , was marked in the Figure S9. It can be observed that La-Cu HS and La-Cu SS show the similar ΔE , which is smaller than that of La₂O₃ and La(OH)₃, suggesting that the La species oxidized state of La-Cu HS and La-Cu SS is distinct from that of La₂O₃ and La(OH)₃.^{34, 35}" Please see them in pages 5-6 of the revised manuscript.

3. La L₃-edge XAS. In the EXAFS curves, the authors claimed the absence of La-La path at 4.0 Å and the presence of La-Cu path at 4.5 Å. This length is more likely a La-O-Cu path. If La and Cu atoms form a bond like the cases in La-Cu alloy or La adatom on Cu surface, the La-Cu bond length should be shorter than 4 Å according to the atomic radii.

Response: We thank the referee again for the comment. The reviewer is right. The peak at 4.5 Å can be assigned to La-O-Cu path, and the presence La of in La-Cu HS and La-Cu SS should be in the form of La-O_x sites on the Cu surface. We have modified the description in the revised manuscript as "One main peak at around 2.0 Å was observed in the FT EXAFS spectra of La-Cu HS and La-Cu SS, which was attributed to La-O coordination. There is no La-La coordination peak at around 4.0 Å in La-Cu HS and La-Cu SS, confirming that the La species exists as single atomic species in La-Cu HS and La-Cu SS without long-range coordination to other La centers. Moreover, there is a peak at around 4.5 Å in La-Cu HS and La-Cu SS FT EXAFS spectra, which could be assigned to the La-O-Cu scattering.³⁶ Therefore, the La species in La-Cu HS and La-Cu SS are the form of La-O_x sites on the Cu surface." Please see them in pages 6-7 of the revised manuscript.

4. Therefore, based on the comments above, in my opinion, a more reasonable structure of the La-Cu catalysts should be La-O_x sites on Cu surface. These sites may be discrete. Thus, their properties may differ from La₂O₃ and La(OH)₃.

But I think metallic La atom sites are unlikely to be formed on Cu. The La-Ox site may also act as a Lewis acid site which can activate CO₂ and promote the CO₂ reduction process. I suggest the authors to conduct DFT simulations based on the model with La-Ox site on Cu surface to check whether this structure favors CO₂ reduction to C₂₊ products

Response: We thank the referee again for the comment. According to the comment, we re-constructed the model with La-O site on Cu surface and conducted DFT simulations. The results indicate that the introduction of La-O site favors CO₂ reduction to C₂₊ products. In the revised manuscript, we have discussed them by “To further verify the abovementioned conclusion, we conducted the first-principles calculation based on density functional theory (DFT). La-doped Cu model (La-Cu) was constructed by introducing La and O atoms into Cu(111) crystal plane to form La-O-Cu sites (Figure S29). In order to investigate the effect of La atom content on the C-C coupling reaction, we introduced 1, 2, 4, and 6 La and O atoms into Cu(111) surface (denoted as 1La-Cu, 2La-Cu, 4La-Cu, 6La-Cu), constructed models with varying La and O atom contents, and calculated the Gibbs free energy change of the 2*CO-to-O*CCO step ($\Delta G_{(2^*CO-to-O^*CCO)}$). Notably, to account for the influence of H₂O and K⁺ on the reaction while simultaneously reducing computational complexity, we simplified the model by placing one K⁺ ion hydrated with six water molecules on the surface of the constructed model (Figures S30-S34).^{21, 46} As shown in Figure 4D, with the increasing the number of La atoms, the $\Delta G_{(2^*CO-to-O^*CCO)}$ gradually decreased. When the number of doped La atoms was 4, the $\Delta G_{(2^*CO-to-O^*CCO)}$ reached its minimum value of 0.17 eV. Further increasing the number of doped La atoms resulted in the increase of $\Delta G_{(2^*CO-to-O^*CCO)}$. The results indicated that moderate La doping was favorable to C-C coupling process, which aligned with the volcano-shaped relationship observed in the experiments between C₂₊ products FE and La doping content. Additionally, we conducted ab initio molecular dynamics (AIMD) simulations over 4La-Cu under 300 K (Figure S35). The results confirmed the robust stability of 4La-Cu configuration, where La and O atoms can exist on Cu(111) surface stably.

Figure 4E shows the Gibbs free energy diagrams for CO₂-to-*OCCO on Cu and 4La-Cu models (Figures S36, S37). The *CO₂ formation was an exothermic process on 4La-Cu, suggesting that the CO₂ molecule was favorable to be adsorbed on 4La-Cu surface. The *OCCO formation from 2*CO was the potential limiting step for the Cu model. However, when 4 La atoms were doped onto the Cu surface, the $\Delta G_{(2^*CO-to-O^*CCO)}$ decreased from 1.39 eV to 0.17 eV, further demonstrating that the La dopants was favorable to C-C coupling process, and thus improving C₂₊ products selectivity. The HER over Cu and 4La-Cu were calculated (Figure S38). The energy barrier over 4La-Cu was 0.36 eV, which was larger than that over Cu (0.31 eV), indicating that the doping of La is beneficial for suppressing HER. Additionally, we adopted the difference between the thermodynamic limiting potentials (ΔU_L) for CO₂-to-*OCCO and HER (*i.e.*, $U_L(\text{CO}_2\text{RR})-U_L(\text{HER})$) to compare the selectivity of CO₂RR and HER,^{47, 48} and more positive ΔU_L corresponds to higher selectivity toward CO₂ reduction. The potential limiting step of CO₂-to-*OCCO over Cu was 2*CO-to-O*CCO step, corresponding to $U_L(\text{CO}_2\text{RR})$ of -1.39 eV, while it was *CO₂-to-

*COOH step over 4La-Cu, corresponding to $U_L(\text{CO}_2\text{RR})$ of -0.90 eV. The $U_L(\text{CO}_2\text{RR})$, $U_L(\text{HER})$ and ΔU_L of 4La-Cu and Cu were displayed in Figure 4F. The results suggested that 4La-Cu was more favorable to undergoing CO_2RR compared to Cu.” Please see them in pages 14-15 of the revised manuscript.

Reviewer 2: The revised version of manuscript NCOMMS-23-44546A incorporates more rigorous DFT calculations, specifically examining solvation effects and doping concentration. However, I am hesitant to fully recommend its publication in Nature Communications unless the authors address the following concerns adequately:

Response: We thank the referee very much for the comment, which help and guide us to improve our work greatly. We have tried our best to answer the questions from the reviewers.

1. I appreciate the authors' consideration of including water and cation in the revised models. However, I question the robustness of the DFT calculations due to the insufficient number of water molecules. It appears that, in some cases, *OCCO is stabilized by the interaction of hydrogen bonding and K^+ -*OCCO, but the K^+ ion is sometimes distant from the adsorbed *CO.

Response: We thank the referee again for the comment. Introducing enough water molecules and potassium ions to create a solvation layer on the surface of the models can make the simulated system closer to real reaction conditions, thus providing precise theoretical explanations for experimental results. However, due to limited computational capacity, it is challenging for us to further increase the number of water molecules and potassium ions. The manuscript reported that combining microenvironment modulation by porous channel structure and intrinsic catalytic activity enhancement via La-O_x doping Cu could promote efficient CO₂RR toward C₂₊ products in strong acidic electrolyte. In CO₂RR performance experiments, we observed that doping La-O_x sites into Cu effectively enhanced the selectivity of C₂₊ products. *In-situ* attenuated total reflection-surface-enhanced IR absorption spectroscopy (ATR-SEIRAS) also confirmed that the doping of La-O_x sites promoted the generation of the crucial intermediate *OCCO involved in the formation of C₂₊ products. The purpose of employing DFT calculations is to assist in validating the results obtained from CO₂RR performance experiments and *in-situ* ATR-SEIRAS experiments. To making the simulation closer to real conditions, we have placed water molecules and potassium ions on the model's surface to the best of our abilities and based on relevant literature (*Nat. Commun.*, 2022, 13, 7596; *Energy Environ. Sci.*, 2024, 17, 510-517). The computational results indicate that the doping of La-O_x into Cu can indeed effectively reduce the Gibbs free energy change of C-C coupling step to generate *OCCO, consistent with the experimental results.

2. The water structure seems distant from the real electrolyte; hence, the authors should consider incorporating more water molecules to establish a continuous hydrogen network.

Response: We thank the referee again for the comment. The main content of our manuscript primarily focuses on experimental work, and the calculations were used to assist in explaining the experimental results. The effects of catalyst composition and structure have been proved through the theoretical calculations. Considering the solvation effect, we also modified the catalyst models by placing one K^+ hydrated with six water molecules on the surface,

which has been described in the few literature (*Nat. Commun.*, 2022, 13, 7596; *Energy Environ. Sci.*, 2024, 17, 510-517). However, it is extremely challenging with our current computational capabilities to further increase the number of water molecules to form a continuous hydrogen bond network and calculate reaction pathways within this network. We agree with the referee that the further theoretical study of solvent effect on the reaction is a very important topic to closely mimic real reaction system in computational simulations, but it is very hard. It needs to enhance computational capabilities and develop advanced computational methods. We believe this is the main reason for that many published papers in CO₂RR field did not consider solvent effect in the DFT computation. Fortunately, it does not influence the conclusion in our paper that La-Cu HS has superior electrochemical performance, because the conclusion is obtained by comparing the performances of different catalysts using the same electrolyte. In the future, we hope that we can carry out the theoretical calculation with the development of related theory. We also greatly appreciate the reviewer for pointing out the improvement direction to our future research.

3. The authors need to clarify the principles used to determine these structures, whether through random sampling or identifying the most thermodynamically stable configuration. Conducting ab initio molecular dynamics simulations for an extended time might help ascertain whether these structures are trapped in local minima.

Response: We thank the referee again for the comment. According to the comment, we have conducted ab initio molecular dynamics simulations over 4La-Cu under 300 K. The results confirmed the robust stability of 4La-Cu configuration, which were shown in Figure S35 in the revised supporting information. In the revised manuscript, we have discussed them by “Additionally, we conducted ab initio molecular dynamics (AIMD) simulations over 4La-Cu under 300 K (Figure S35), the results confirmed the robust stability of 4La-Cu configuration, where La and O atoms can exist on Cu(111) surface stably.” Please see them in page 15 of the revised manuscript.

Meanwhile, the description of ab initio molecular dynamics simulations has been added in the Method section of the revised manuscript as “To obtain the temperature and energy state, the constrained ab initio molecular dynamics (AIMD) has been used with slow-growth method. At the beginning of MD simulation, the models have been heated up to 300 K by velocity scaling over 1.5 ps and then equilibrated at 300 K for 5 ps with a 2-fs time step.” Please see them in page 19 of the revised manuscript.

4. It is crucial for the authors to consider the corresponding initial and final states of 2*CO and *OCCO. The identified 2*CO structures under different concentrations might not be the most thermodynamically favored ones.

Response: We thank the referee again for the comment. We can answer the question from two aspects. On the one hand, the stable structure of 2*CO and *OCCO can be obtained on the models with different *CO concentrations during the computational process. On the other hand, we observed the *CO and *OCCO intermediates in the *in-situ* ATR-SEIRAS spectra, demonstrating that *CO and *OCCO remain stable in the reaction process. Therefore, we

can confirm that the ^{13}C -OCCO was formed through the C-C coupling of 2^{13}C O.

Reviewer 3: All my concerns have been addressed. We recommend the acceptance of the manuscript by Nature Communications.

Response: We thank the reviewer very much for the positive comment.

REVIEWERS' COMMENTS

Reviewer #1 (Remarks to the Author):

All my concerns have been addressed. I recommend this article to be accepted.

Reviewer #2 (Remarks to the Author):

The authors have addressed all my comments and thus I am happy to recommend its acceptance for publication on nature communications.

Responses to the comments of the reviewers

Reviewer 1:

All my concerns have been addressed. I recommend this article to be accepted.

Response: We thank the reviewer very much for the positive comment.

Reviewer 2:

The authors have addressed all my comments and thus I am happy to recommend its acceptance for publication on nature communications.

Response: We thank the reviewer very much for the positive comment.